# ENERGY-WEIGHTED FLOW MATCHING FOR OFFLINE REINFORCEMENT LEARNING

**Shiyuan Zhang**[1]  **Weitong Zhang**[2]  **Quanquan Gu**[3]
[1]Tsinghua University  [2]UNC-Chapel Hill  [3]University of California, Los Angeles
`shiyuan-21@mails.tsinghua.edu.cn, weitongz@unc.edu, qgu@cs.ucla.edu`

## ABSTRACT

This paper investigates energy guidance in generative modeling, where the target distribution is defined as $q(\mathbf{x}) \propto p(\mathbf{x}) \exp(-\beta \mathcal{E}(\mathbf{x}))$, with $p(\mathbf{x})$ being the data distribution and $\mathcal{E}(\mathbf{x})$ as the energy function. To comply with energy guidance, existing methods often require auxiliary procedures to learn intermediate guidance during the diffusion process. To overcome this limitation, we explore energy-guided flow matching, a generalized form of the diffusion process. We introduce energy-weighted flow matching (EFM), a method that directly learns the energy-guided flow without the need for auxiliary models. Theoretical analysis shows that energy-weighted flow matching accurately captures the guided flow. Additionally, we extend this methodology to energy-weighted diffusion models and apply it to offline reinforcement learning (RL) by proposing the Q-weighted Iterative Policy Optimization (QIPO). Empirically, we demonstrate that the proposed QIPO algorithm improves performance in offline RL tasks. Notably, our algorithm is the first energy-guided diffusion model that operates independently of auxiliary models and the first exact energy-guided flow matching model in the literature.

## 1 INTRODUCTION

Recent years have witnessed the success of applying diffusion models (Ho et al., 2020; Song et al., 2020) and flow matching models (Chen et al., 2018; Lipman et al., 2022) to generative models. Given this success, another important aspect is to *guide* generative models to achieve specific, controlled outputs, such as generating images for a certain class (Ho & Salimans, 2021; Dhariwal & Nichol, 2021), designing molecular structures with desired properties (Wang et al., 2024; Hoogeboom et al., 2022), or improving policies for reinforcement learning (Wang et al., 2022; Lu et al., 2023). Guidance can come from various sources, such as classifiers, including both classifier guidance (Dhariwal & Nichol, 2021) and classifier-free guidance (Ho & Salimans, 2021). In addition, Lu et al. (2023) proposed using guidance from an *energy function*, where the distribution is generated from $q(\mathbf{x}) \propto p(\mathbf{x}) \exp(-\beta \mathcal{E}(\mathbf{x}))$, where the model is guided to generate data $\mathbf{x}$ with lower energy $\mathcal{E}(\mathbf{x})$ from the original data distribution.

Several recent efforts have been made to learn and sample from the guided distribution $q(\mathbf{x})$ using diffusion models. Chen et al. (2022) performed rejection sampling from the learned data distribution $p(\mathbf{x})$. Lu et al. (2023) introduced an *intermediate energy function* $\mathcal{E}_t(\cdot)$, allowing the score function of $q_t(\mathbf{x})$ to be decomposed as $\nabla_{\mathbf{x}} \log q_t(\mathbf{x}) = \nabla_{\mathbf{x}} \log p_t(\mathbf{x}) - \nabla_{\mathbf{x}} \mathcal{E}_t(\mathbf{x})$ within the diffusion process. Lu et al. (2023) also proposed contrastive energy prediction for training the intermediate energy $\mathcal{E}_t$, relying on back-propagation to calculate its gradient with respect to $\mathbf{x}$. Wang et al. (2024) proposed to directly approximate the gradient of this intermediate energy function $\mathcal{E}_t$ as the 'force-field' guidance. However, all these methods require either additional neural networks, back-propagation, or post-processing to compose the guided distribution $q(\mathbf{x})$, which introduces unnecessary errors and complexity. Therefore, the following question arises:

> ***Q1.*** *Can we directly obtain an energy-guided diffusion model without auxiliary models?*

Another challenge for energy-guided generative models lies in providing guidance in flow matching models (Chen et al., 2018; Lipman et al., 2022), which is a more general, simulation-free counterpart to diffusion models. Zheng et al. (2023) explored the use of classifier-free guidance for flow

matching in offline RL. However, since flow matching models approximate the velocity field $\mathbf{u}_t(\mathbf{x})$ for the dynamics of the probability density path $p_t(\mathbf{x})$, it is highly non-trivial to obtain the guided velocity field $\widehat{\mathbf{u}}_t(\mathbf{x})$ for the distribution $q_t(\mathbf{x})$ under energy guidance. This presents the second key question:

*Q2. Can we inject exact energy guidance into general flow matching models?*

In this paper, we answer the aforementioned two questions affirmatively by proposing an energy-guided velocity field and an *energy-weighted* flow matching objective, with extensions to *energy-weighted* diffusion models and applications in offline reinforcement learning. Our contributions are summarized as follows:

- In response to **Q2.**, for general flow matching, we propose the energy-guided velocity field $\widehat{\mathbf{u}}_t(\mathbf{x})$, based on the conditional velocity field $\mathbf{u}_{t0}(\mathbf{x}|\mathbf{x}_0)$. The proposed $\widehat{\mathbf{u}}_t(\mathbf{x})$ is theoretically guaranteed to generate the energy-guided distribution $q(\mathbf{x}) \propto p(\mathbf{x}) \exp(-\beta \mathcal{E}(\mathbf{x}))$.
- We introduce the *energy-weighted* flow matching loss to train a neural network $\mathbf{v}_t^{\boldsymbol{\theta}}$ that approximates $\widehat{\mathbf{u}}_t(\mathbf{x})$. The energy-weighted flow matching only requires the conditional vector field $\mathbf{u}_t(\mathbf{x}|\mathbf{x}_0)$ and the energy $\mathcal{E}(\mathbf{x}_0)$ for $\mathbf{x}_0$ from the dataset. As the answer to **Q1.**, we extend this approach to diffusion models, proposing the energy-weighted diffusion model. Energy-weighted diffusion model learns an energy-guided diffusion model directly without any auxiliary model.
- We apply these methods to offline reinforcement learning tasks to evaluate the performance of the energy-weighted flow matching and diffusion models. Under this framework, we introduce an *iterative policy refinement* technique for offline reinforcement learning. Empirically, we demonstrate that the proposed method achieves superior performance across various offline RL tasks.

**Notations.** Vectors are denoted by lowercase boldface letters, such as $\mathbf{x}$, and matrices by uppercase boldface letters, such as $\mathbf{A}$. For any positive integer $k$, the set $1, 2, \ldots, k$ is denoted by $[k]$, and we define $\overline{[k]} = [k] \cup \{0\}$. The natural logarithm of $x$ is denoted by $\log x$. We use $p_t$ to represent the marginal distribution of $\mathbf{x}$ at time $t$, and $p_{0t}$ to represent the conditional distribution of $\mathbf{x}_0$ given $\mathbf{x}_t$. Similarly, $p_0$ denotes the original data distribution for the diffusion model at $t = 0$, while $p_{t0}$ represents the conditional distribution of $\mathbf{x}_t$ given $\mathbf{x}_0$ in the forward process of the diffusion model.

## 2 RELATED WORK

**Diffusion Models and Flow Matching Models.** Diffusion models (Ho et al., 2020) and score matching (Song et al., 2020) have emerged as powerful generative modeling techniques in tasks such as image synthesis (Dhariwal & Nichol, 2021), text-to-image generation (Podell et al., 2023), and video generation (Ho et al., 2022). In addition to these frontier applications, the success of diffusion models has been further enhanced by accelerated sampling processes (Lu et al., 2022a;b) and the extension of diffusion models to discrete value spaces (Austin et al., 2021). Alongside the success of diffusion models, flow matching models (Lipman et al., 2022; Chen et al., 2018) provide an alternative for simulation-free generation. Unlike score-based approaches, flow matching models aim to learn the velocity field that transports data points from the initial noise distribution to the target data distribution. This velocity field can be viewed as a generalized form of the reverse process in diffusion models and can be extended to optimal transport (Villani et al., 2009), rectified flow (Liu, 2022), and more complex flows.

**Guidance in Diffusion and Flow Matching Models.** Beyond learning and generating the original data distribution with diffusion or flow matching models, significant efforts have been made to control the generation process to produce data with specific desired properties. Dhariwal & Nichol (2021) introduced classifier guidance, which decomposes the conditional score function $\nabla \log p(\mathbf{x}|y)$ into the sum of the data distribution gradient $\nabla \log p(\mathbf{x})$ and the gradient from a classifier $\nabla \log p(y|\mathbf{x})$. To simplify this, Ho & Salimans (2021) proposed classifier-free guidance, which directly integrates $\nabla \log p(y|\mathbf{x})$ into the score function. Sendera et al. (2024) studied diffusion-structured samplers by introducing the inductive bias in Langevin process. Lu et al. (2023); Chen et al. (2022) further explored energy-based guidance, where the target distribution is defined as $q(\mathbf{x}) \propto p(\mathbf{x}) \exp(-\beta \mathcal{E}(\mathbf{x}))$. Unlike classifier guidance, energy-based guidance extends to real-valued energy functions $\mathcal{E}$, making it particularly relevant for tasks such as molecular structure generation. Specifically, Chen et al. (2022); Cremer et al. (2024) employed rejection sampling to implement energy guidance, while Lu et al. (2023); Wang et al. (2024) used auxiliary models to esti-

Table 1: Comparison between guidance methods. *Exact Guidance?* means if the model can generate $p(\mathbf{x})p^\beta(c|\mathbf{x})$ when $\beta \neq 1$. *w/o Auxiliary Model?* means if the method can direct learn the guidance without auxiliary model ($\checkmark$) or not ($\times$).

| Guidance | Exact Guidance? | w/o Auxiliary Model? |
|---|---|---|
| Classifier-guidance (Dhariwal & Nichol, 2021) | $\times$ | $\times$ |
| Classifier-free guidance (Ho & Salimans, 2021) | $\times$ | $\checkmark$ |
| Contrastive energy prediction (Lu et al., 2023) | $\checkmark$ | $\times$ |
| **Energy-weighted diffusion (ours)** | $\checkmark$ | $\checkmark$ |

mate the guidance from the energy function. We defer a more formal, technical comparison between the energy-based guidance and classifier-based guidance in Table 1 in Section 4.3. In the context of flow matching, Zheng et al. (2023) introduced classifier-free guidance for flow matching in the domain of offline reinforcement learning.

**Diffusion and Flow Matching Models in Reinforcement Learning.** Recent advances in diffusion models and flow matching models have enabled a range of applications in reinforcement learning (RL). Janner et al. (2022); Wang et al. (2022) explore modeling behavior policies using diffusion models. Building on these results, Chen et al. (2022); Lu et al. (2023) model the offline RL objective as an energy-guided diffusion process, while Ajay et al. (2022); Zheng et al. (2023) apply the same policy optimization using classifier-free diffusion and flow matching models. Chen et al. (2023); Hansen-Estruch et al. (2023) use diffusion models to regularize the distance between the optimal policy and the behavioral policy, and Fang & Lan (2024); He et al. (2023) leverage diffusion models for constrained policy optimization. Another line of research (Jackson et al., 2024; Lee et al., 2024; Lu et al., 2024) focuses on using generative models to augment synthetic datasets.

## 3 PRELIMINARIES

### 3.1 CONDITIONAL FLOW MATCHING FOR GENERATIVE MODELING

Continuous Normalizing Flows (CNFs) (Chen et al., 2018) considers the dynamic of the probability density function by *probability density path* $p : [0, 1] \times \mathbb{R}^d \mapsto \mathbb{R}_{\geq 0}$ which transmits between the data distribution $p_0$ and the initial distribution (e.g., Gaussian distribution) $p_1$. The *flow* $\phi : [0, 1] \times \mathbb{R}^d \mapsto \mathbb{R}^d$ is constructed by a *vector field* $\mathbf{v} : [0, 1] \times \mathbb{R}^d \mapsto \mathbb{R}^d$ describing the velocity of the particle at position $\mathbf{x}$, i.e., $\frac{\mathrm{d}}{\mathrm{d}t}\phi_t(\mathbf{x}) = \mathbf{v}_t(\phi_t(\mathbf{x}))$ where $\phi_1(\mathbf{x}) = \mathbf{x}$. [1]

In order to ensure that the vector field $\mathbf{v}$ *generates* the probability density path $p_t$, the following *continuity equation* (Villani et al., 2009) is required:

$$\frac{\mathrm{d}}{\mathrm{d}t}p_t(\mathbf{x}) + \mathrm{div} \cdot [p_t(\mathbf{x})\mathbf{v}_t(\mathbf{x})] = 0, \quad \forall \mathbf{x} \in \mathbb{R}^d. \tag{3.1}$$

The objective of flow matching is to learn a neural network $\mathbf{v}_t^{\boldsymbol{\theta}}$ to learn the ground truth vector field $\mathbf{u}_t$ by minimizing their differences, i.e., $\mathcal{L}_{\mathrm{FM}}(\boldsymbol{\theta}) = \mathbb{E}_{t,p_t(\mathbf{x})}\|\mathbf{v}_t^{\boldsymbol{\theta}}(\mathbf{x}) - \mathbf{u}_t(\mathbf{x})\|_2^2$ with respect to the network parameter $\boldsymbol{\theta}$. However, it is infeasible to calculate the ground truth vector field $\mathbf{u}_t$. To address this issue, Lipman et al. (2022) suggests to match the *conditional vector field* $\mathbf{u}_{t0}(\mathbf{x}|\mathbf{x}_0)$ instead of the vector field $\mathbf{u}_t(\mathbf{x})$, as presented by the following theorem:

**Theorem 3.1** (Theorem 1, 2; Lipman et al. 2022). Given the conditional vector field $\mathbf{u}_{t0}(\mathbf{x}|\mathbf{x}_0)$ that generates the conditional distribution $p_{t0}(\mathbf{x}|\mathbf{x}_0)$, then the "marginal" vector field $\mathbf{u}_t(\mathbf{x}) = \int_{\mathbf{x}_0} p_{0t}(\mathbf{x}_0|\mathbf{x})\mathbf{u}_{t0}(\mathbf{x}|\mathbf{x}_0)\mathrm{d}\mathbf{x}_0$ generates the marginal distribution $p_t(\mathbf{x})$. In addition, up to a constant factor independent of $\boldsymbol{\theta}$, *Flow Matching* loss $\mathcal{L}_{\mathrm{FM}}(\boldsymbol{\theta})$ and *Conditional Flow Matching* loss $\mathcal{L}_{\mathrm{CFM}}(\boldsymbol{\theta})$ are equal, where

$$\mathcal{L}_{\mathrm{FM}}(\boldsymbol{\theta}) = \mathbb{E}_{t,\mathbf{x}}\|\mathbf{v}_t^{\boldsymbol{\theta}}(\mathbf{x}) - \mathbf{u}_t(\mathbf{x})\|_2^2, \ \mathcal{L}_{\mathrm{CFM}}(\boldsymbol{\theta}) = \mathbb{E}_{t,\mathbf{x}_0,\mathbf{x}}\|\mathbf{v}_t^{\boldsymbol{\theta}}(\mathbf{x}) - \mathbf{u}_{t0}(\mathbf{x}|\mathbf{x}_0)\|_2^2, \tag{3.2}$$

where $t \sim \lambda(t)$, $\mathbf{x}_0$ follows the data distribution $p_0(\cdot)$ and $\mathbf{x} \sim p_{t0}(\cdot|\mathbf{x}_0)$ where $p_{t0}$ is generated by conditional vector field $\mathbf{u}_{t0}$. Hence $\nabla_{\boldsymbol{\theta}}\mathcal{L}_{\mathrm{FM}}(\boldsymbol{\theta}) = \nabla_{\boldsymbol{\theta}}\mathcal{L}_{\mathrm{CFM}}(\boldsymbol{\theta})$.

---

[1] We adapt the notation of diffusion to unify the diffusion and flow matching. The notation here is different from flow matching notations in Chen et al. (2018); Lipman et al. (2022), where $p_1$ represent the data distribution. The flow matching result remains unchanged except switching $t = 1$ with $t = 0$.

In practice, the conditional distribution $p_{t0}(\mathbf{x}|\mathbf{x}_0)$ is usually modeled as a *Gaussian path* with $p_{t0}(\mathbf{x}|\mathbf{x}_0) = \mathcal{N}(\mu_t \mathbf{x}_0, \sigma_t^2 \mathbf{I})$. Zheng et al. (2023) suggests that in this case, conditional flow matching is equivalent to the score matching (Song et al., 2020):

**Lemma 3.2** (Lemma 1, Zheng et al. 2023). Let $p_{t0}(\mathbf{x}|\mathbf{x}_0)$ be a Gaussian path with scheduler $(\mu_t, \sigma_t)$, i.e., $p_{t0}(\mathbf{x}|\mathbf{x}_0) = \mathcal{N}(\mu_t \mathbf{x}_0, \sigma_t^2 \mathbf{I})$, then the velocity field $\mathbf{u}_{t0}(\mathbf{x}|\mathbf{x}_0)$ is related to the score function $\nabla_{\mathbf{x}} \log p_{t0}(\mathbf{x}|\mathbf{x}_0)$ by

$$\mathbf{u}_{t0}(\mathbf{x}|\mathbf{x}_1) = \dot{\mu}_t \mu_t^{-1} \mathbf{x} + (\dot{\mu}_t \sigma_t - \mu_t \dot{\sigma}_t) \sigma_t \mu_t^{-1} \nabla_{\mathbf{x}} \log p_{t0}(\mathbf{x}|\mathbf{x}_0), \qquad (3.3)$$

where $\dot{\mu}_t$ and $\dot{\sigma}_t$ are both the derivative of $\mu_t$ and $\sigma_t$ with respect to time $t$.

In addition, Zheng et al. (2023) proved that the reverse process of this diffusion process with Gaussian path $\mu_t, \sigma_t$ can be written by

$$\frac{\mathrm{d}\mathbf{x}}{\mathrm{d}t} = \frac{\dot{\mu}_t}{\mu_t} \mathbf{x} + (\dot{\mu}_t \sigma_t - \mu_t \dot{\sigma}_t) \frac{\sigma_t}{\mu_t} \nabla_{\mathbf{x}} \log p_t(\mathbf{x}) = \mathbf{u}_t(\mathbf{x}). \qquad (3.4)$$

## 3.2 ENERGY-GUIDED DIFFUSION MODELS

The standard diffusion model aims to learn and generate from the data distribution $p_0$. However, instead of generating from $p_0$, there are a series of applications consider sampling from an energy-guided distribution $q_0(\mathbf{x}) \propto p_0(\mathbf{x}) \exp(-\beta \mathcal{E}(\mathbf{x}))$ where $\mathcal{E} : \mathbb{R}^d \mapsto \mathbb{R}$ is the energy function and $\beta \in \mathbb{R}^+$ is the strength of the guidance. Lu et al. (2023) suggested to construct the score function $\nabla_{\mathbf{x}} \log q_t(\mathbf{x})$ from the original score function $\nabla_{\mathbf{x}} \log p_t(\mathbf{x})$ by introducing the *intermediate energy function* $\mathcal{E}_t(\mathbf{x})$ through the following theorem:

**Theorem 3.3** (Theorem 3.1, Lu et al. (2023)[2]). Let $q_0(\mathbf{x}) \propto p_0(\mathbf{x}) \exp(-\beta \mathcal{E}(\mathbf{x}))$ and define the forward process as $q_{t0}(\mathbf{x}|\mathbf{x}_0) = p_{t0}(\mathbf{x}|\mathbf{x}_0) = \mathcal{N}(\mu_t \mathbf{x}_0, \sigma_t^2 \mathbf{I})$, and the marginal distribution $q_t(\mathbf{x}), p_t(\mathbf{x})$ at time $t$ defined by

$$q_t(\mathbf{x}) = \int_{\mathbf{x}_0} q_{t0}(\mathbf{x}|\mathbf{x}_0) q_0(\mathbf{x}_0) \mathrm{d}\mathbf{x}_0, \qquad p_t(\mathbf{x}) = \int_{\mathbf{x}_0} p_{t0}(\mathbf{x}|\mathbf{x}_0) p_0(\mathbf{x}_0) \mathrm{d}\mathbf{x}_0.$$

Let the intermediate energy function be

$$\mathcal{E}_t(\mathbf{x}) = - \log \mathbb{E}_{p_{0t}(\mathbf{x}_0|\mathbf{x})}[\exp(-\beta \mathcal{E}(\mathbf{x}_0)], \qquad (3.5)$$

then the marginal distribution $p_t$ and $q_t$ satisfy

$$q_t(\mathbf{x}) \propto p_t(\mathbf{x}) \exp(-\mathcal{E}_t(\mathbf{x})), \qquad \nabla_{\mathbf{x}} \log q_t(\mathbf{x}) = \nabla_{\mathbf{x}} \log p_t(\mathbf{x}) - \nabla_{\mathbf{x}} \mathcal{E}_t(\mathbf{x}). \qquad (3.6)$$

Therefore, Lu et al. (2023) suggests to firstly learn the intermediate energy function $\mathcal{E}_t$ using *contrastive energy prediction* (CEP) and to learn the score function $\nabla_{\mathbf{x}} \log p_t(\mathbf{x})$ using standard diffusion models (e.g., DDPM (Ho et al., 2020)). Then the score function of the energy-guided distribution $\nabla_{\mathbf{x}} \log q_t(\mathbf{x})$ can therefore be composed according to (3.6).

## 4 METHODOLOGY

In this section, we propose a *energy-weighted* method for training both CNFs and diffusion models to generate the energy-guided distribution $q(\mathbf{x}) \propto p(\mathbf{x}) \exp(-\beta \mathcal{E}(\mathbf{x}))$. Compared with Lu et al. (2023); Wang et al. (2024), the energy-weighted method provide a more straightforward way to obtain the energy-guided generative models and removes the necessaries of estimating the intermediate energy function $\mathcal{E}_t(\mathbf{x})$ and its gradient $\nabla_{\mathbf{x}} \mathcal{E}_t(\mathbf{x})$.

### 4.1 ENERGY-WEIGHTED FLOW MATCHING

In this subsection, we construct a new energy guided flow to generate the energy-guided probability distribution. We also proposed two equivalent loss function to train the neural networks for approximating the energy-guided flow. We start by the first theorem suggesting a energy-guided flow to generate the energy-guided probability distribution $q_t(\mathbf{x}) \propto p_t(\mathbf{x}) \exp(-\mathcal{E}_t(\mathbf{x}))$.

---

[2]We swap the notation $p$ and $q$ to align with our notation systems.

**Theorem 4.1.** Given an energy function $\mathcal{E}(\cdot)$ and a conditional flow $\mathbf{u}_{t0}(\mathbf{x}|\mathbf{x}_0)$ that generates the probability distribution $p_{t0}(\mathbf{x}|\mathbf{x}_0)$, the energy guided distribution $q_t(\mathbf{x}) \propto p_t(\mathbf{x}) \exp(-\mathcal{E}_t(\mathbf{x}))$ is generated by the flow

$$\widehat{\mathbf{u}}_t(\mathbf{x}) = \int_{\mathbf{x}_0} p_{0t}(\mathbf{x}_0|\mathbf{x})\mathbf{u}_t(\mathbf{x}|\mathbf{x}_0)\frac{\exp(-\beta\mathcal{E}(\mathbf{x}_0))}{\exp(-\mathcal{E}_t(\mathbf{x}))}\mathrm{d}\mathbf{x}_0, \tag{4.1}$$

which will generate distribution $q_0(\mathbf{x}) \propto p_0(\mathbf{x}) \exp(-\beta\mathcal{E}(\mathbf{x}))$. The intermediate energy function is defined in (3.5).

**Remark 4.2.** Theorem 4.1 suggests a method to construct the vector field $\widehat{\mathbf{u}}_t(\mathbf{x})$ from the conditional vector field $\mathbf{u}_{t0}(\mathbf{x}|\mathbf{x}_0)$ and the intermediate energy function $\mathcal{E}_t(\mathbf{x})$ in the closed-form solution. It holds universally to any conditional flow including the the optimal transport, Gaussian path or rectify flow. We will extend the discussion on the diffusion models in the next subsection.

Despite the closed-form expression for the energy-guided flow, it remains challenging to learn a neural network $\mathbf{v}_t^{\boldsymbol{\theta}}$ to match $\widehat{\mathbf{u}}_t$ since the following two reasons. First, $\widehat{\mathbf{u}}_t$ in (4.1) requires to sample over data distribution $\mathbf{x}_0$. Secondly, the expression of $\widehat{\mathbf{u}}_t$ still requires the estimation of the intermediate energy function $\mathcal{E}_t$. Previous methods are both using auxiliary neural networks to approximate either $\mathcal{E}_t$ (Lu et al., 2023) or its gradient $\nabla_{\mathbf{x}}\mathcal{E}_t(\mathbf{x})$ (Wang et al., 2024). To overcome these two challenges, the following theorem suggests a *weighted* flow matching objective which can be directly used to learn $\widehat{\mathbf{u}}_t$ without the aforementioned procedures.

**Theorem 4.3.** Define the **Energy-weighted Flow Matching** loss $\mathcal{L}_{\mathrm{EFM}}$ as

$$\mathcal{L}_{\mathrm{EFM}}(\boldsymbol{\theta}) = \mathbb{E}_{t,\mathbf{x}}\left[\frac{\exp(-\mathcal{E}_t(\mathbf{x}))}{\mathbb{E}_{p_t(\widetilde{\mathbf{x}})}[\exp(-\mathcal{E}_t(\widetilde{\mathbf{x}}))]}\|\mathbf{v}_t^{\boldsymbol{\theta}}(\mathbf{x}) - \widehat{\mathbf{u}}_t(\mathbf{x})\|_2^2\right], \tag{4.2}$$

and the **Conditional Energy-weighted Flow Matching** loss $\mathcal{L}_{\mathrm{CEFM}}$ as

$$\mathcal{L}_{\mathrm{CEFM}}(\boldsymbol{\theta}) = \mathbb{E}_{t,\mathbf{x},\mathbf{x}_0}\left[\frac{\exp(-\beta\mathcal{E}(\mathbf{x}_0))}{\mathbb{E}_{p_0(\widetilde{\mathbf{x}_0})}[\exp(-\beta\mathcal{E}(\widetilde{\mathbf{x}_0}))]}\|\mathbf{v}_t^{\boldsymbol{\theta}}(\mathbf{x}) - \mathbf{u}_{t0}(\mathbf{x}|\mathbf{x}_0)\|_2^2\right], \tag{4.3}$$

where the expectation on $t$ is taken over some predefined distribution $\lambda(t)$, $\mathbf{x}_0$ is sampled from the data distribution $p_0(\cdot)$ and $\mathbf{x}$ at time $t$ is sampled by $p_t(\mathbf{x})$ with conditional distribution $p_{t0}(\mathbf{x}|\mathbf{x}_0)$ generated by the flow $\mathbf{u}_{t0}(\mathbf{x}|\mathbf{x}_0)$. $\mathcal{L}_{\mathrm{EFM}}(\boldsymbol{\theta})$ and $\mathcal{L}_{\mathrm{CEFM}}(\boldsymbol{\theta})$ are equal up to a constant factor. Hence $\nabla_{\boldsymbol{\theta}}\mathcal{L}_{\mathrm{EFM}}(\boldsymbol{\theta}) = \nabla_{\boldsymbol{\theta}}\mathcal{L}_{\mathrm{CEFM}}(\boldsymbol{\theta})$.

Theorem 4.3 suggests that minimizing $\mathcal{L}_{\mathrm{CEFM}}$ is equivalent to minimizing $\mathcal{L}_{\mathrm{EFM}}$. It is obvious that the global minimum of $\mathcal{L}_{\mathrm{CEFM}}$ is $\mathbf{v}_t^{\boldsymbol{\theta}}(\mathbf{x}) = \widehat{\mathbf{u}}_t(\mathbf{x})$, given enough neural network capacity and infinite data. Therefore, one can use $\mathcal{L}_{\mathrm{CEFM}}$ to directly learn the guided flow $\widehat{\mathbf{u}}_t(\mathbf{x})$, without calculating the intermediate energy function $\mathcal{E}_t(\mathbf{x})$ or its gradient.

Besides the aforementioned message, Theorem 4.3 suggests several understandings and intuitions in training the neural network $\mathbf{v}_t^{\boldsymbol{\theta}}$ which are discussed as follows

**Remark 4.4** (Regarding the weighted energy guided loss $\mathcal{L}_{\mathrm{EFM}}$). Instead of directly minimizing $\mathbb{E}_{t,\mathbf{x}}\|\mathbf{v}_t^{\boldsymbol{\theta}}(\mathbf{x}) - \widehat{\mathbf{u}}_t(\mathbf{x})\|_2^2$, $\mathcal{L}_{\mathrm{EFM}}$ places higher weight on the input $\mathbf{x}$ with a lower intermediate energy $\mathcal{E}_t(\mathbf{x})$. Intuitively speaking, $\exp(-\mathcal{E}(\mathbf{x}))$ can be viewed as a prior distribution in generating $q_t(\mathbf{x}) \propto p_t(\mathbf{x})\exp(-\mathcal{E}_t(\mathbf{x}))$. Therefore, for all time $t$, areas with higher $\exp(-\mathcal{E}(\mathbf{x}))$ will be more likely to be visited. As a result, it would be more efficient placing more importance on $\mathbf{x}$ in these areas instead of learning $\widehat{\mathbf{u}}_t(\mathbf{x})$ uniformly for all $\mathbf{x} \in \mathbb{R}^d$.

**Remark 4.5** (Regarding the conditional weighted energy guided loss $\mathcal{L}_{\mathrm{CEFM}}$). The weight $\exp(-\beta\mathcal{E}(\mathbf{x}_0))$ suggests how the energy "guides" the conditional flow matching. Fixing $t$ and $\mathbf{x}$ and changing the form of expectations in (4.3), $\mathcal{L}_{\mathrm{CEFM}}(\boldsymbol{\theta})$ becomes

$$\mathcal{L}_{\mathrm{CEFM}}(\boldsymbol{\theta}; t, \mathbf{x}) = \mathbb{E}_{p_{0t}(\mathbf{x}_0|\mathbf{x})}\left[\frac{\exp(-\beta\mathcal{E}(\mathbf{x}_0))}{\mathbb{E}_{p_0(\widetilde{\mathbf{x}_0})}[\exp(-\beta\mathcal{E}(\widetilde{\mathbf{x}_0}))]}\|\mathbf{v}_t^{\boldsymbol{\theta}}(\mathbf{x}) - \mathbf{u}_{t0}(\mathbf{x}|\mathbf{x}_0)\|_2^2\right].$$

Intuitively speaking, velocity field $\mathbf{u}_{t0}(\mathbf{x}|\mathbf{x}_0)$ will move the particle $\mathbf{x}$ to $\mathbf{x}_0$. Therefore, when the energy guidance does not exist (i.e, $\mathcal{E}(\mathbf{x}) = 0$), $\mathbf{v}_t^{\boldsymbol{\theta}}(\mathbf{x})$ is essentially finding the "center" of all $\mathbf{x}_0$ possibly generated from $\mathbf{x}$ following $p(\mathbf{x}_0|\mathbf{x})$. In the presence of the energy function $\mathcal{E}(\mathbf{x}_0)$, the learnt vector field $\mathbf{v}_t^{\boldsymbol{\theta}}(\mathbf{x})$ is biased to the conditional vector field $\mathbf{u}_{t0}(\mathbf{x}|\mathbf{x}_0)$ with higher weight $\exp(-\beta\mathcal{E}(\mathbf{x}_0))$. As a result, the learnt velocity field $\mathbf{v}_t^{\boldsymbol{\theta}}(\mathbf{x})$ will generate $\mathbf{x}_0$ with lower energy $\mathcal{E}(\mathbf{x}_0)$.

**Remark 4.6** (Connection with the importance sampling). The conditional weighted energy guided loss $\mathcal{L}_{\text{CEFM}}$ can be also interpreted from the importance sampling techniques. Suppose we can sample directly from the data $q_0(\mathbf{x}) \propto p_0(\mathbf{x}) \exp(-\beta\mathcal{E}(\mathbf{x}))$, minimizing the following loss $\mathcal{L}_q$ will get a velocity field $\mathbf{v}_t$ for generating distribution $q_0$

$$\mathcal{L}_q(\theta) = \mathbb{E}_{t,\mathbf{x}_0 \sim q_0(\mathbf{x}), \mathbf{x} \sim q_{t0}(\mathbf{x}|\mathbf{x}_0)}[\|\mathbf{v}_t^{\theta}(\mathbf{x}) - \mathbf{u}_{t0}(\mathbf{x}|\mathbf{x}_0)\|_2^2],$$

where $q_{t0}(\mathbf{x}|\mathbf{x}_0) = p_{t0}(\mathbf{x}|\mathbf{x}_0)$. Since , where $Z$ is a constant, changing the data distribution from $q_0$ to $p_0$ yields that

$$\mathcal{L}_q(\theta) = \mathbb{E}_{t,\mathbf{x}_0 \sim p_0(\mathbf{x}), \mathbf{x} \sim q_{t0}(\mathbf{x}|\mathbf{x}_0)} \left[ \frac{q_0(\mathbf{x})}{p_0(\mathbf{x})} \|\mathbf{v}_t^{\theta}(\mathbf{x}) - \mathbf{u}_{t0}(\mathbf{x}|\mathbf{x}_0)\|_2^2 \right]$$

$$= \mathbb{E}_{t,\mathbf{x}_0 \sim p_0(\mathbf{x}), \mathbf{x} \sim p_{t0}(\mathbf{x}|\mathbf{x}_0)} \left[ \frac{\exp(-\beta\mathcal{E}(\mathbf{x}_0))}{\mathbb{E}_{p_0(\widetilde{\mathbf{x}}_0)}[\exp(-\beta\mathcal{E}(\widetilde{\mathbf{x}}_0))]} \|\mathbf{v}_t^{\theta}(\mathbf{x}) - \mathbf{u}_{t0}(\mathbf{x}|\mathbf{x}_0)\|_2^2 \right] = \mathcal{L}_{\text{CEFM}}(\theta),$$

where the second equation is given by $q_0(\mathbf{x}) = p_0(\mathbf{x}) \exp(-\beta\mathcal{E}(\mathbf{x}))/\mathbb{E}_{\mathbf{x}_0}[\exp(-\beta\mathcal{E}(\mathbf{x}_0))]$ according to Lemma B.1.

## 4.2 WEIGHTED DIFFUSION MODELS

Theorem 4.3 suggests a general method to learn an energy-guided flow $\mathbf{v}^{\theta}$ given any condition flow $\mathbf{u}_{t0}(\mathbf{x}|\mathbf{x}_0)$, including diffusion flow (Song et al., 2020), optimal transport (Lipman et al., 2022), rectified flow (Liu, 2022) or even more complicated $\mathbf{u}_{t0}(\mathbf{x}|\mathbf{x}_0)$. In this subsection, we restrict the analysis to the diffusion flow and present several useful analysis for the diffusion and score matching models. The first corollary provides the closed-form score function $\nabla_{\mathbf{x}} \log q_t(\mathbf{x})$ for the energy-guided distribution $q_t(\mathbf{x}) \propto p_t(\mathbf{x}) \exp(-\mathcal{E}_t(\mathbf{x}))$:

**Corollary 4.7.** Under the assumptions claimed in Lemma 3.2, when $p_{t0}(\mathbf{x}|\mathbf{x}_0)$ is a Gaussian path with scheduler $(\mu_t, \sigma_t)$, we have $\nabla_{\mathbf{x}} \log q_t(\mathbf{x}) = \nabla_{\mathbf{x}} \log p_t(\mathbf{x}) - \nabla_{\mathbf{x}}\mathcal{E}_t(\mathbf{x})$ and

$$\nabla_{\mathbf{x}} \log q_t(\mathbf{x}) = \int_{\mathbf{x}_0} p_{0t}(\mathbf{x}_0|\mathbf{x}) \nabla_{\mathbf{x}} \log p_{t0}(\mathbf{x}|\mathbf{x}_0) \frac{\exp(-\beta\mathcal{E}(\mathbf{x}_0))}{\exp(-\mathcal{E}_t(\mathbf{x}))} d\mathbf{x}_0, \tag{4.4}$$

where $\nabla_{\mathbf{x}} \log p_{t0}(\mathbf{x}|\mathbf{x}_0) = -(\mathbf{x} - \mu_t\mathbf{x}_0)/\sigma_t^2 = -\boldsymbol{\epsilon}/\sigma_t, \boldsymbol{\epsilon} \sim \mathcal{N}(0, \mathbf{I}_d)$.

Corollary 4.7 suggests a method to estimate the guided score function without calculating the gradient of the intermediate energy function $\nabla_{\mathbf{x}}\mathcal{E}_t(\mathbf{x})$ as conducted in Lu et al. (2023). Then the following corollary suggests a similar energy-weighted diffusion model to train this score function $\nabla_{\mathbf{x}} \log q_t(\mathbf{x})$ in practice.

**Corollary 4.8.** Define the **E**nergy-weighted **D**iffusion loss $\mathcal{L}_{\text{ED}}$ and the **C**onditional **E**nergy-weighted **D**iffusion loss $\mathcal{L}_{\text{CED}}$ separately as

$$\mathcal{L}_{\text{ED}}(\boldsymbol{\theta}) = \mathbb{E}_{t,\mathbf{x}} \left[ \frac{\exp(-\mathcal{E}_t(\mathbf{x}))}{\mathbb{E}_{p_t(\widetilde{\mathbf{x}})}[\exp(-\mathcal{E}_t(\widetilde{\mathbf{x}}))]} \|\mathbf{s}_t^{\boldsymbol{\theta}}(\mathbf{x}) - \nabla_{\mathbf{x}} \log q_t(\mathbf{x})\|_2^2 \right],$$

$$\mathcal{L}_{\text{CED}}(\boldsymbol{\theta}) = \mathbb{E}_{t,\mathbf{x},\mathbf{x}_0} \left[ \frac{\exp(-\beta\mathcal{E}(\mathbf{x}_0))}{\mathbb{E}_{p_0(\widetilde{\mathbf{x}_0})}[\exp(-\beta\mathcal{E}(\widetilde{\mathbf{x}_0}))]} \left\| \mathbf{s}_t^{\boldsymbol{\theta}}(\mathbf{x}) - \nabla_{\mathbf{x}} \log p_{t0}(\mathbf{x}|\mathbf{x}_0) \right\|_2^2 \right],$$

where the expectation is taken from $t \sim \lambda(t), \mathbf{x}_0 \sim p_0(\mathbf{x}_0)$ and $\mathbf{x} \sim p_{t0}(\mathbf{x}|\mathbf{x}_0)$. Thus the marginal distribution of $\mathbf{x}$ is is $p_t(\mathbf{x})$. $\mathcal{L}_{\text{ED}}(\boldsymbol{\theta})$ is equal with $\mathcal{L}_{\text{CED}}(\boldsymbol{\theta})$ up to a constant and thus $\nabla_{\boldsymbol{\theta}}\mathcal{L}_{\text{ED}}(\boldsymbol{\theta}) = \nabla_{\boldsymbol{\theta}}\mathcal{L}_{\text{CED}}(\boldsymbol{\theta})$.

**Remark 4.9.** A similar approach is proposed in Wang et al. (2024) for estimating $\nabla_{\mathbf{x}}\mathcal{E}_t(\mathbf{x})$ using a neural network by

$$\nabla_{\mathbf{x}}\mathcal{E}_t(\mathbf{x}) = \frac{\mathbb{E}_{p_{0t}(\mathbf{x}_0|\mathbf{x})} \left[ \exp(-\beta\mathcal{E}(\mathbf{x}_0)) \left(\nabla_{\mathbf{x}} \log p_t(\mathbf{x}) - \nabla_{\mathbf{x}} \log p_{t0}(\mathbf{x}|\mathbf{x}_0)\right) \right]}{\exp(-\mathcal{E}_t(\mathbf{x}))}, \tag{4.5}$$

and then plugging (4.5) back to $\nabla_{\mathbf{x}} \log q_t(\mathbf{x}) = \nabla_{\mathbf{x}} \log p_t(\mathbf{x}) - \nabla_{\mathbf{x}}\mathcal{E}_t(\mathbf{x})$ for generating the score function $\nabla_{\mathbf{x}} \log q_t(\mathbf{x})$. However, in order to obtain $\nabla_{\mathbf{x}}\mathcal{E}_t(\mathbf{x})$, Wang et al. (2024) requires to estimate $\exp(\mathcal{E}_t(\mathbf{x}))$ by sampling and approximate $\nabla_{\mathbf{x}}\mathcal{E}_t(\mathbf{x})$ via another neural network. In contrast, using $\mathcal{L}_{\text{CED}}$ as discussed in Corollary 4.8 removes the necessity of estimating both $\nabla_{\mathbf{x}}\mathcal{E}_t(\mathbf{x})$ and $\mathcal{E}_t(\mathbf{x})$. Therefore, Energy-weighted diffusion can directly obtain the score function for guided distribution without additional sampling or back-propagation.

---

**Algorithm 1** Training Energy-Weighted Diffusion Model

---

**Input:** Score function $\mathbf{s}_t^{\boldsymbol{\theta}}(\cdot)$, schedule $(\mu_t, \sigma_t)$, guidance scale $\beta$, batch size $B$, time weight $\lambda(t)$

1: **for** batch $\{\mathbf{x}_0^i, \mathcal{E}(\mathbf{x}_0^i)\}_i$ **do**
2:     **for** index $i \in [B]$ **do**
3:         Calculate guidance $g_i = \text{softmax}(-\beta\mathcal{E}(\mathbf{x}_0^i)) = \exp(-\beta\mathcal{E}(\mathbf{x}_0^i))/\sum_j \exp(-\beta\mathcal{E}(\mathbf{x}_0^j))$
4:         Sample $t_i \sim U(0,1)$, calculate $\mu_{t_i}, \sigma_{t_i}$, sample $\boldsymbol{\epsilon}_i \sim \mathcal{N}(0, \mathbf{I}_d)$ and $\mathbf{x}_{t_i} = \mu_{t_i}\mathbf{x}_0^i + \sigma_{t_i}\boldsymbol{\epsilon}_i$
5:     **end for**
6:     Calculate and take a gradient step using $\mathcal{L}_{\text{CED}}(\boldsymbol{\theta}) = \sum_i \lambda(t_i)g_i\|\mathbf{s}_{t_i}^{\boldsymbol{\theta}}(\mathbf{x}_{t_i}) + \boldsymbol{\epsilon}_i/\sigma_{t_i}\|_2^2$.
7: **end for**

---

In the implementation of the diffusion models, since $\mathbf{x} \sim \mathcal{N}(\mu_t\mathbf{x}_0, \sigma_t^2\mathbf{I})$, the conditional score function $\nabla_{\mathbf{x}} \log p_{t0}(\mathbf{x}|\mathbf{x}_0) = -\boldsymbol{\epsilon}/\sigma_t$ where $\boldsymbol{\epsilon} = (\mathbf{x}_t - \mu\mathbf{x}_0)/\sigma_t \sim \mathcal{N}(\mathbf{0}, \mathbf{I})$. In addition, the denominator $\mathbb{E}_{p_0(\mathbf{x}_0)}[\exp(-\beta\mathcal{E}(\mathbf{x}_0))]$ can be approximated by the empirical average $\sum_{i \in N} \exp(-\beta\mathcal{E}(\mathbf{x}_0^i))/N$ in a batch. A practical algorithm is presented in Algorithm 1. The training process is similar with the standard DDPM (Ho et al., 2020) or score matching (Song et al., 2020). The only difference is that we incorporate the weight by calculating a energy guidance $g_i$ using the softmax value of using $-\beta\mathcal{E}(\mathbf{x}_0^i)$ from the current batch in Line 3.

## 4.3 COMPARISON BETWEEN CEP AND CLASSIFIER (FREE) GUIDANCE

In this subsection, we compare our method with Contrastive Energy Prediction (CEP, Lu et al. 2023), Classifier-Guidance (CG, Dhariwal & Nichol 2021) and Classifier-Free Guidance (CFG, Ho & Salimans 2021). We consider the guided distribution $q_0(\mathbf{x}) \propto p_0(\mathbf{x})p^\beta(c|\mathbf{x})$ where $p_0(c|\mathbf{x})$ is the classifier, $\beta$ is the guidance scale, $c$ is the desired class which we fix during the analysis. Comparing with the formulation of the energy guidance $q_1(\mathbf{x}) \propto p_1(\mathbf{x}) \exp(-\beta\mathcal{E}(\mathbf{x}))$, the "energy function" can be interpreted as $\mathcal{E}(\mathbf{x}) = -\log p(c|\mathbf{x})$. To begin with, the following lemma provides the closed-form solution for the energy-guided diffusion and classifier-guided diffusion

**Lemma 4.10.** Given the same guidance scale $\beta$ and the same diffusion process, let the energy function be defined by $\mathcal{E}(\mathbf{x}) = -\log p(c|\mathbf{x})$, the score function for CG and CFG are both:

$$\nabla_{\mathbf{x}} \log q_t(\mathbf{x}) = \nabla_{\mathbf{x}} \log p_t(\mathbf{x}) + \nabla_{\mathbf{x}} \log \left[\mathbb{E}_{p_{0t}(\mathbf{x}_0|\mathbf{x})}p(c|\mathbf{x}_0)\right]^\beta, \quad (4.6)$$

while the score function for energy-weighted diffusion and CEP are both

$$\nabla_{\mathbf{x}} \log q_t(\mathbf{x}) = \nabla_{\mathbf{x}} \log p_t(\mathbf{x}) + \nabla_{\mathbf{x}} \log \mathbb{E}_{p_{0t}(\mathbf{x}_0|\mathbf{x})}p^\beta(c|\mathbf{x}_0). \quad (4.7)$$

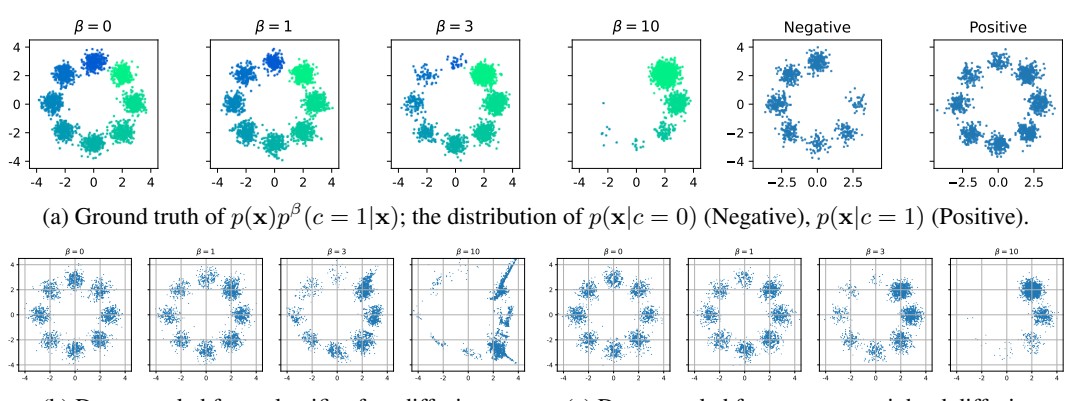

(a) Ground truth of $p(\mathbf{x})p^\beta(c = 1|\mathbf{x})$; the distribution of $p(\mathbf{x}|c = 0)$ (Negative), $p(\mathbf{x}|c = 1)$ (Positive).

(b) Data sampled from classifier-free diffusion.      (c) Data sampled from energy-weighted diffusion.

Figure 1: Visualization of the ground-truth distribution $p(\mathbf{x})p^\beta(c = 1|\mathbf{x})$ with different values of $\beta$, the posterior distribution $p(\mathbf{x}|c)$ with $c \in \{0, 1\}$, and the data sampled from classifier-free diffusion and energy-weighted diffusion. The energy-weighted diffusion process demonstrates better performance when $\beta > 1$. More examples and details of this experiments are provided in Appendix C.

The score function (4.7) is exactly the score function that generates $q_0(\mathbf{x}) \propto p_0(\mathbf{x}) \exp(-\beta\mathcal{E}(\mathbf{x})) = p_0(\mathbf{x})p(c|\mathbf{x})^\beta$ according to Theorem 3.3. In a sharp contrast, (4.6) is not guaranteed to generate

the desired distribution $q_0$ when $\beta \neq 1$ because $[\mathbb{E}_{p_{0t}(\mathbf{x}_0|\mathbf{x})}p(c|\mathbf{x}_0)]^{\beta} \neq \mathbb{E}_{p_{0t}(\mathbf{x}_0|\mathbf{x})}p^{\beta}(c|\mathbf{x}_0)$. As demonstrated in Figure 1, when $\beta = 1$, the distributions generated by CFG and energy-guided diffusion are both similar to the ground-truth distribution. However, when $\beta > 1$, the distribution generated by CFG differs from the ground-truth distribution, whereas the energy-guided diffusion can still generate the ground-truth distribution. Finally, the following lemma also verifies that when $\beta = 1$, the energy-weighted diffusion process is the same with the classifier-free guidance to learn the score function of the posterior distribution $\nabla_{\mathbf{x}} \log p_t(\mathbf{x}|c)$:

**Lemma 4.11.** Let $\beta = 1$ and assume that $\mathcal{E}(\mathbf{x}) = -\log p(c|\mathbf{x})$ for some fixed $c$, then $\mathcal{L}_{\mathrm{CED}}(\boldsymbol{\theta})$ is

$$\mathcal{L}_{\mathrm{CED}}(\boldsymbol{\theta}) = \mathbb{E}_{t,\mathbf{x},\mathbf{x}_0|c}\|\mathbf{s}_t^{\boldsymbol{\theta}}(\mathbf{x}) - \nabla_{\mathbf{x}} \log p_{t0}(\mathbf{x}|\mathbf{x}_0)\|_2^2, \tag{4.8}$$

where the expectation is taken over $t \sim \lambda(t)$, $\mathbf{x}_0 \sim p_0(\cdot|c)$ and $\mathbf{x} \sim p_{t0}(\cdot|\mathbf{x}_0)$. The global optimal for (4.8) is $\mathbf{s}_t^{\boldsymbol{\theta}}(\mathbf{x}) = \nabla_{\mathbf{x}} \log p_t(\mathbf{x}|c)$.

In addition to the *exact* guidance provided by the CEP and energy-weighted diffusion, it is important to highlight that the energy-weighted diffusion model eliminates the necessity for the intermediate energy model. This advantage is similar to the simplicity provided by the CFG, compared with CG. We summarized the difference and connection between energy-weighted diffusion, CEP, CFG and CG in Table 1.

# 5 Q-WEIGHTED ITERATIVE POLICY OPTIMIZATION FOR OFFLINE RL

We consider the episodic Markov Decision Processes denoted by $\mathcal{M}(\mathcal{S}, \mathcal{A}, r, \mathbb{P}, \gamma)$ with $\mathcal{S}, \mathcal{A}$ denoting the state and action space respectively. $r$ is the reward function, $\mathbb{P}(\cdot|\mathbf{s}, \mathbf{a})$ is the transition kernel, and $\gamma$ is the discount factor. In offline RL, the data is collected by a *behavioral policy* $\mu$. The policy optimization with KL regularization (Peters et al., 2010; Peng et al., 2019) is formulated as

$$\underset{\pi^{\boldsymbol{\theta}}}{\arg\max} \mathbb{E}_{\mathbf{a} \sim \pi^{\boldsymbol{\theta}}(\cdot|\mathbf{x})} Q(\mathbf{x}, \mathbf{a}) - \tfrac{1}{\beta}\mathrm{KL}(\pi^{\boldsymbol{\theta}} \| \mu), \tag{5.1}$$

where $\mathbf{x}$ denotes the state and $\mathbf{a}$ denotes the action. $Q(\mathbf{x}, \mathbf{a})$ is the estimation of the state-action value function $Q^{\pi}(\mathbf{x}, \mathbf{a}) = \mathbb{E}[\sum_{t=0}^{\infty} \gamma_t r(\mathbf{x}_t, \mathbf{a}_t)|\mathbf{x}_0 = xb, \mathbf{a}_0 = \mathbf{a}, \pi]$. The closed-form solution to (5.1) is $\pi(\mathbf{a}|\mathbf{x}) \propto \mu(\mathbf{a}|\mathbf{x}) \exp(\beta Q(\mathbf{x}, \mathbf{a}))$. Following the procedure of the energy-weighted diffusion model discussed in Section 4, we propose learning the score function $\mathbf{s}_t^{\boldsymbol{\theta}}(\mathbf{a}, \mathbf{x})$ or the velocity field $\mathbf{v}_t^{\boldsymbol{\theta}}(\mathbf{a}, \mathbf{x})$ that generates the optimal policy $\pi(\cdot|\mathbf{x})$ using the Q-weighted diffusion loss $\mathcal{L}_{\mathrm{QD}}$ or Q-weighted flow matching loss $\mathcal{L}_{\mathrm{QF}}$, respectively, defined as:

$$\mathcal{L}_{\mathrm{QD}}(\boldsymbol{\theta}) = \mathbb{E}_{t,(\mathbf{x},\mathbf{a}),\mathbf{a}_t}\left[\frac{\exp(\beta Q(\mathbf{x}, \mathbf{a}))}{\mathbb{E}_{\widetilde{\mathbf{a}} \sim \mu(\cdot|\mathbf{x})}[\exp(\beta Q(\mathbf{x}, \widetilde{\mathbf{a}}))]}\left\|\mathbf{s}_t^{\boldsymbol{\theta}}(\mathbf{a}_t; \mathbf{x}) - \nabla_{\mathbf{a}_t} \log p_{t0}(\mathbf{a}_t|\mathbf{a}_0)\right\|_2^2\right], \tag{5.2}$$

$$\mathcal{L}_{\mathrm{QF}}(\boldsymbol{\theta}) = \mathbb{E}_{t,(\mathbf{x},\mathbf{a}),\mathbf{a}_t}\left[\frac{\exp(\beta Q(\mathbf{x}, \mathbf{a}))}{\mathbb{E}_{\widetilde{\mathbf{a}} \sim \mu(\cdot|\mathbf{x})}[\exp(\beta Q(\mathbf{x}, \widetilde{\mathbf{a}}))]}\left\|\mathbf{v}_t^{\boldsymbol{\theta}}(\mathbf{a}_t; \mathbf{x}) - \mathbf{u}_{t0}(\mathbf{a}_t|\mathbf{a})\right\|_2^2\right], \tag{5.3}$$

where the expectation is taken over $t \sim \lambda(t)$, $(\mathbf{x}, \mathbf{a})$ is sampled from the offline RL dataset, and $\mathbf{a}_t \sim p_{t0}(\cdot|\mathbf{a})$. Two components are essential for training either (5.2) or (5.3): First, the behavioral policy $\mu(\cdot|\mathbf{x})$ can be trained via standard diffusion or flow matching models. Second, any $Q$ function derived from offline RL algorithms can be used as $Q(\mathbf{x}, \mathbf{a})$ in (5.2) or (5.3).

## 5.1 PROPOSED ALGORITHM

We present the algorithm sketch for the $Q$-weighted diffusion process in Algorithm 2, and defer the detailed implementation to Appendix D. From a high-level overview, the algorithm first trains the score model to match the behavioral policy $\mu(\mathbf{a}, \mathbf{x})$ for $K_1$ episodes, then trains the $Q$ function for $K_2$ episodes. The algorithm then performs the *Q-weighted iterative policy optimization* (QIPO) as follows: First, in Line 10, using the current score function $\mathbf{s}^{\boldsymbol{\theta}}$, the algorithm samples several support actions $\mathbf{a}_{ij}$ to estimate the expectation $\mathbb{E}_{\widetilde{\mathbf{a}} \sim \mu(\cdot|\mathbf{x})}$ in (5.2). In Line 15, the algorithm optimizes $\boldsymbol{\theta}$ with respect to the empirical estimation of $\mathcal{L}_{\mathrm{QD}}(\boldsymbol{\theta})$. Therefore, assuming the score function $\mathbf{s}_t^{\boldsymbol{\theta}}(\cdot|\mathbf{x})$ generates the behavior distribution $\mu(\cdot|\mathbf{x})$ after the warm-up for $K_1$ episodes, using the sampled support action $\mathbf{a}_{ij}$, the optimal score function for Line 15 corresponds to the policy $\pi_1 = \mu(\cdot|\mathbf{x}) \exp(\beta Q^{\psi}(\cdot|\mathbf{x}))$. When the number of episodes $k = K_{\mathrm{renew}} + 1$, the algorithm revisits Line 10 with the new score function and regenerates the support action set with the new policy

---

**Algorithm 2** Q-weighted iterative policy optimization for offline RL (diffusion)

---

**Input:** Score function $\mathbf{s}_t^{\boldsymbol{\theta}}(\cdot)$, schedule $(\mu_t, \sigma_t)$, guidance scale $\beta$, batch size $B$, weight on time $\lambda(t)$
**Input:** Offline RL dataset $\mathcal{D} = \{(\mathbf{x}, \mathbf{a}, \mathbf{x}', r)\}$, number of epochs $K_1, K_2$ and $K_3$, Q function $Q^{\boldsymbol{\psi}}$
**Input:** Support action set size $M$, support action set renew frequency $K_{\text{renew}}$

1: **for** diffusion model warm-up epoch $k \in [K_1]$ **do**
2:      Train $\mathbf{s}_t^{\boldsymbol{\theta}}$ for each batch $\{(\mathbf{x}_i, \mathbf{a}_i, \mathbf{x}_i', r_i)\}_i \subset \mathcal{D}$ following standard diffusion model.
3: **end for**
4: **for** Q-learning warm-up epoch $k \in [K_2]$ **do**
5:      Train $Q^{\boldsymbol{\psi}}$ for each batch $\{(\mathbf{x}_i, \mathbf{a}_i, \mathbf{x}_i', r_i)\}_i \subset \mathcal{D}$
6: **end for**
7: **for** policy improvement step $k \in [K_3]$ **do**
8:      **for** batch $\{(\mathbf{x}_i, \mathbf{a}_i, \mathbf{x}_i', r_i)\}_i \subset \mathcal{D}$ **do**
9:          **if** $k \bmod K_{\text{renew}} = 1$ **then**
10:              Sample support action set $\mathbf{a}_{ij}$ using score function $\mathbf{s}^{\boldsymbol{\theta}}(\cdot|\mathbf{x}_i)$ for all $i \in [B], j \in [M]$
11:              Denote $\mathbf{a}_{i0} = \mathbf{a}_i$, sample $t_{ij} \sim U(0,1)$ and $\mathbf{a}_{ij,t_{ij}} \sim \mathcal{N}(\mu_{t_{ij}} \mathbf{a}_{ij}, \sigma_{t_{ij}}^2 \mathbf{I})$ for all $j \in \overline{[M]}$
12:              Calculate guidance $g_{ij} = \text{softmax}_j(\beta Q^{\boldsymbol{\psi}}(\mathbf{x}_i, \mathbf{a}_{ij})) = \frac{\exp(\beta Q^{\boldsymbol{\psi}}(\mathbf{x}_i, \mathbf{a}_{ij}))}{\sum_{j=0}^M \exp(\beta Q^{\boldsymbol{\psi}}(\mathbf{x}_i, \mathbf{a}_{ij}))}$
13:          **end if**
14:          Calculate loss $\mathcal{L}_{\text{QD}}(\boldsymbol{\theta}) = \sum_{i=1,j=0}^{B,M} \lambda(t_{ij}) g_{ij} \big\| \mathbf{s}_{t_{ij}}^{\boldsymbol{\theta}}(\mathbf{a}_{ij,t_{ij}}; \mathbf{x}_i) - \nabla_{\mathbf{a}_t} \log p_{t0}(\mathbf{a}_{ij,t_{ij}} | \mathbf{a}_{ij}) \big\|_2^2$
15:          Update $\boldsymbol{\theta}$ using the gradient of $\mathcal{L}_{\text{QD}}(\boldsymbol{\theta})$
16:      **end for**
17: **end for**

---

$\pi_1$. Thus the target score function for Line 15 to optimize is $\pi_2(\cdot|\mathbf{x}) \propto \pi_1(\cdot|\mathbf{x}) \exp(\beta Q^{\boldsymbol{\psi}}(\cdot|\mathbf{x})) \propto \mu(\cdot|\mathbf{x}) \exp(2\beta Q^{\boldsymbol{\psi}}(\cdot|\mathbf{x}))$. As a result, denoting $l = (k-1) \bmod K_{\text{renew}}$, the policy $\pi_l$ generated by the score function $\mathbf{s}_t^{\boldsymbol{\theta}_k}$ is:

$$\pi_{l+1}(\mathbf{a}|\mathbf{x}) \propto \pi_l(\mathbf{a}|\mathbf{x}) \exp(\beta Q^{\boldsymbol{\psi}}(\mathbf{a},\mathbf{x})) \propto \cdots \propto \mu(\mathbf{a}|\mathbf{x}) \exp((l+1)\beta Q^{\boldsymbol{\psi}}(\mathbf{a},\mathbf{x})), \quad (5.4)$$

which will implicitly increase the guidance scale $\beta$.

Similar weighted policy optimization approaches have been applied in Kang et al. (2024); Ding et al. (2024). However, QIPO builds the relationship between the KL-regularized policy optimization so that QIPO can iteratively improve the policy as described in (5.4). Compared with directly setting a large guidance scale $\beta$, QIPO makes the support action set $\widetilde{a}$ to be more concentrated in the space with higher $Q$ values. As a result, QIPO learns a more robust Q-weighted score function $\mathbf{s}_t^{\boldsymbol{\theta}}$ compared to one-step Q-weighted diffusion with a larger $\beta$. Second, QGPO (Lu et al., 2023) introduces a scaling factor $s$ and composes the score function as $\nabla_{\mathbf{a}_t} \log \pi_t(\mathbf{a}_t|\mathbf{x}) = \nabla_{\mathbf{a}_t} \log \mu_t(\mathbf{a}_t|\mathbf{x}) + s \nabla_{\mathbf{a}_t} Q_t(\mathbf{x}, \mathbf{a}_t)$, where $Q_t$ is the intermediate $Q$ function, similar to the $\mathcal{E}_t$ in Section 4. However, as we discussed in Section 4.3 about the comparison of the CFG, since

$$sQ_t(\mathbf{a}_t, \mathbf{x}) = -s \log \mathbb{E}_{p_{0t}(\mathbf{a}|\mathbf{a}_t)}[\exp(\beta Q(\mathbf{a}, \mathbf{x})] \neq \log \mathbb{E}_{p_{0t}(\mathbf{a}|\mathbf{a}_t)}[\exp(s\beta Q(\mathbf{a}, \mathbf{x})],$$

using a guidance scale $s > 1$ does not guarantee generating a policy strictly regularized by the behavioral policy $\mu(\mathbf{a}|\mathbf{x})$. In contrast, as (5.4) suggests, our approach strictly follows the formulation $\pi(\mathbf{a}|\mathbf{x}) \propto \mu(\mathbf{a}|\mathbf{x}) \exp(s\beta Q^{\boldsymbol{\psi}}(\mathbf{a}|\mathbf{x}))$ regularized by the behavioral policy.

## 5.2 EXPERIMENT RESULTS

We evaluate the performance of QIPO with flow matching and diffusion model on the D4RL tasks (Fu et al., 2020) in this subsection.

**Experiment configurations** We implement the flow matching model **QIPO-OT** using the optimal-transport conditional velocity fields (Lipman et al., 2022) and the diffusion model **QIPO-Diff** using VP-SDE (Song et al., 2020). We use the same network structure as QGPO for a fair comparison of the efficiency with QGPO. We defer the detailed experiment configurations in Appendix E.1.

We compare our results with other state-of-the-art benchmarks, including Diffusion-QL (Wang et al., 2022), QGPO (Lu et al., 2023), IDQL (Hansen-Estruch et al., 2023), SRPO (Chen et al., 2023) and Guided Flows (Zheng et al., 2023) and present the results in Table 2. As the experiment results

Table 2: Evaluation numbers of D4RL benchmarks (normalized as suggested by Fu et al. (2020)). We report mean ± standard deviation of algorithm performance across 8 random seeds. The highest performance is **boldfaced highlighted.** The performance within 5% of the maximum absolute value in every individual task are highlighted.

| Dataset | Environment | SfBC | QGPO | IDQL | SRPO | Guided Flows | QIPO-Diff (ours) | QIPO-OT (ours) |
|---|---|---|---|---|---|---|---|---|
| Medium-Expert | HalfCheetah | 92.6 | 93.5 | 95.9 | 92.2 | **97** | $94.14 \pm 0.48$ | $94.45 \pm 0.49$ |
| Medium-Expert | Hopper | 108.6 | 108.0 | 108.6 | 100.1 | 105 | $\mathbf{112.12 \pm 0.42}$ | $108.02 \pm 5.19$ |
| Medium-Expert | Walker2d | 109.8 | 110.7 | 112.7 | **114.0** | 94 | $110.14 \pm 0.51$ | $110.87 \pm 1.04$ |
| Medium | HalfCheetah | 45.9 | 54.1 | 51.0 | **60.4** | 49 | $48.19 \pm 0.20$ | $54.16 \pm 1.27$ |
| Medium | Hopper | 57.1 | **98.0** | 65.4 | 95.5 | 84 | $89.53 \pm 9.96$ | $94.05 \pm 13.27$ |
| Medium | Walker2d | 77.9 | 86.0 | 82.5 | 84.4 | 77 | $84.99 \pm 0.46$ | $\mathbf{87.61 \pm 1.46}$ |
| Medium-Replay | HalfCheetah | 37.1 | 47.6 | 45.9 | **51.4** | 42 | $45.27 \pm 0.42$ | $48.04 \pm 0.79$ |
| Medium-Replay | Hopper | 86.2 | 96.9 | 92.1 | 101.2 | 89 | $101.23 \pm 0.47$ | $\mathbf{101.25 \pm 2.18}$ |
| Medium-Replay | Walker2d | 65.1 | 84.4 | 85.1 | 84.6 | 78 | $90.08 \pm 4.53$ | $78.57 \pm 26.09$ |
| **Average (Locomotion)** | | 75.6 | 86.6 | 82.1 | 87.1 | 79.4 | 86.2 | 86.3 |
| Default | AntMaze-umaze | 92.0 | 96.4 | 94.0 | 97.1 | - | $\mathbf{97.5 \pm 0.53}$ | $93.62 \pm 7.05$ |
| Diverse | AntMaze-umaze | **85.3** | 74.4 | 80.2 | 82.1 | - | $73.88 \pm 6.42$ | $76.12 \pm 9.93$ |
| Play | AntMaze-medium | 81.3 | 83.6 | **84.5** | 80.7 | - | $82.75 \pm 3.24$ | $80.00 \pm 13.66$ |
| Diverse | AntMaze-medium | 82.0 | 83.8 | 84.8 | 75.0 | - | $86.00 \pm 8.65$ | $\mathbf{86.42 \pm 5.44}$ |
| Play | AntMaze-large | 59.3 | 66.6 | 63.5 | 53.6 | - | $\mathbf{73.25 \pm 10.90}$ | $55.5 \pm 29.39$ |
| Diverse | AntMaze-large | 45.5 | 64.8 | **67.9** | 53.6 | - | $40.5 \pm 20.40$ | $32.13 \pm 23.16$ |
| **Average (AntMaze)** | | 74.2 | 78.3 | **79.1** | 73.6 | - | 77.3 | 71.96 |

suggests, QIPO-Diff and QIPO-OT consistently outperform the baselines in various tasks. We defer more baseline algorithms for comparison to Table 4 in Appendix E.

Among these benchmark algorithms, we would like to highlight the comparison between *Guided Flows* (Zheng et al., 2023) and *Q-Guided Policy Optimization* (QGPO, Lu et al. 2023). Firstly, compared with Guided Flows (Zheng et al., 2023), QIPO-OT enjoys higher performance across many different tasks. This improved performance is due to the fact that energy-based guidance will provide more accurate guidance compared with classifier-free guidance, as discussed in Section 4.3.

Secondly, compared with QGPO (Lu et al., 2023), QIPO-Diff directly learns the energy-guided score function without estimating the intermediate energy function. As a result, QIPO does not require the back-propagation to calculate the gradient of the intermediate energy function $\nabla_{\mathbf{x}}\mathcal{E}_t(\mathbf{x})$ and therefore enjoys a faster sampling speed compared with QGPO when using the same score network, as presented in Table 3. In addition, since the QIPO guarantees the strict formulation $\pi(\mathbf{a}|\mathbf{x}) \propto \mu(\mathbf{a}|\mathbf{x})\exp(s\beta Q^{\psi}(\mathbf{a}|\mathbf{x}))$ as shown in (5.4), QIPO enjoys better performance compared with QGPO on various tasks. We defer more detailed discussions on the advantage of QIPO-OT to Appendix E.2.

Table 3: Comparison of the running time for action generation between QGPO (Lu et al., 2023) and QIPO, averaged over 1500 runs. The percentage reduction in time compared to QGPO is also reported.

| Method | Time (ms) |
|---|---|
| QGPO (Lu et al., 2023) | 75.05 |
| **QIPO-OT (ours)** | **27.26** (-63.68%) |
| **QIPO-Diff (ours)** | 55.86 (-25.57%) |

**Ablation Study.** We conduct ablation study on changing the support action set $M$, policy renew period $K_{\text{renew}}$ and the guidance scale $\beta$. We defer the detailed ablation study result in Appendix E.3.

## 6 CONCLUSION AND FUTURE WORK

In this paper, we explored the energy guidance in both flow matching and diffusion models. We introduced Energy-weighted Flow Matching (EFM) and Energy-weighted Diffusion (ED) by incorporating the energy guidance directly into these generative models without relying on auxiliary models or post-processing steps. We applied the proposed methods in offline RL and introduced Q-weighted Iterative Policy Optimization (QIPO), which enjoys competitive empirical performance on the D4RL benchmark. To the best of our knowledge, this work is the first to present an energy-guided flow matching model and the first algorithm to *directly* learn an energy-guided diffusion model. While the current QIPO focuses on offline RL without interacting with the environment, we leave the extension to online RL, where guidance from the $Q$-function can be updated through online interactions

## ETHICS STATEMENT

This paper focuses on the methodology of generative models and strictly adheres to ethical guidelines and standards. As with other deep generative modeling techniques, energy-weighted diffusion and flow matching model have the potential to generate harmful content, such as "deepfakes", and may reflect or amplify unwanted social biases present in the training data. However, these broader societal impacts are not directly relevant to the specific contributions of our work. Therefore, we do not believe any unique ethical concerns or negative social impacts arise from this paper.

## REPRODUCIBILITY STATEMENT

We provide detailed descriptions of our experimental setups, dataset construction processes, and code implementation in the supplementary materials to ensure the reproducibility of the Energy-weighted Diffusion and QIPO methods. The full experimental configurations are presented in Appendix E.1. To further support research in the community, we will release the model checkpoints following the de-anonymization process.

## ACKNOWLEDGMENTS

We thank the anonymous reviewers for their helpful comments. Part of this work was done while WZ was a PhD student at UCLA. WZ and QG are supported in part by the NSF grants DMS-2323113, CPS-2312094, IIS-2403400 and the research fund from UCLA-Amazon Science Hub. WZ is also supported by UCLA dissertation year fellowship. Part of this work is supported by the Google Cloud Research Credits program with the award GCP376319164. The views and conclusions contained in this paper are those of the authors and should not be interpreted as representing any funding agencies.

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

## A  COMPARISON WITH OTHER DIFFUSION-BASED RL

Besides the aforementioned works on RL, there is another series of work on using diffusion models in RL by considering the reward-weighted regression (Peters & Schaal, 2007) where the score-matching loss function formulates $\mathbb{E}[f(Q(s,a))\|\epsilon_{\boldsymbol{\theta}}(a';s,t)-\epsilon)\|_2^2]$. In particular, EDP (Kang et al., 2024) considers the $f(Q(s,a))$ in the format of CRR or IQL formulated as $f_{\text{CRR}} = \exp[(Q_\phi(s,a) - \mathbb{E}_{a'\sim\widehat{\pi}(a'|s)}Q(s,a'))/\tau_{\text{CRR}}]$ or $f_{\text{IQL}} = \exp[(Q_\phi(s,a) - V_\psi(s))/\tau_{\text{IQL}}]$ and QVPO (Ding et al., 2024) considers a more aggressive exploration in online RL with $f(Q(s,a)) = Q(s,a)$. Compared with QIPO, $f_{\text{CRR}}$ assumes $\widehat{\pi}(a'|s) = \mathcal{N}(\widehat{a}_0|s)$ with $\widehat{a}_0 = \sqrt{1/\bar{\alpha}^k}a^k - \sqrt{1/\bar{\alpha}^k - 1}\epsilon_{\boldsymbol{\theta}}(a^k, k; s)$. This oversimplifies the policy as a Gaussian policy and requires the score-network $\epsilon_{\boldsymbol{\theta}}$ for calculate the weight. $f_{\text{IQL}}$ requires an additional value network for $V_\psi(s)$. Most importantly, both EDP (Kang et al., 2024) and QVPO (Ding et al., 2024), are designed to optimize the diffusion policy to maximize the $Q$ function. However, in the energy-guided generation for $\pi(a|s) \propto \mu(a|s)\exp(Q(s,a))$, where both maximizing the $Q$ value and aligning with the $Q$ function are equally important, these works cannot align with the original distribution $\mu$ and therefore cannot be applied to guide broader generative models applications like image synthesis or molecular conformation generation.

## B  THEORETICAL PROOF OF THE RESULTS

We provide the theoretical proof for all theorems, corollaries and lemmas in this section.

### B.1  PROOF OF THEOREM 4.1

We start by the following lemma which provides the closed-form expression of the distribution $q_t(\mathbf{x}) \propto p_t(\mathbf{x})\exp(-\mathcal{E}_t(\mathbf{x}))$

**Lemma B.1.** The closed-form expression of $q_t(\mathbf{x}) \propto p_t(\mathbf{x})\exp(-\mathcal{E}_t(\mathbf{x}))$ is

$$q_t(\mathbf{x}) = \frac{p_t(\mathbf{x})\exp(-\mathcal{E}_t(\mathbf{x}))}{\mathbb{E}_{\mathbf{x}_0}[\exp(-\beta\mathcal{E}(\mathbf{x}_0))]}, \tag{B.1}$$

where the expectation is taken with respect to the data distribution $p_0(\mathbf{x}_0)$. The denominator $\mathbb{E}_{\mathbf{x}_0}[\exp(-\beta\mathcal{E}(\mathbf{x}_0))]$ is a constant with respect to $\mathbf{x}$ and $t$.

*Proof.* It is easy to verify that

$$\begin{aligned}
\int_{\mathbf{x}} p_t(\mathbf{x})\exp(-\mathcal{E}_t(\mathbf{x}))\mathrm{d}\mathbf{x} &= \int_{\mathbf{x}} p_t(\mathbf{x})\int_{\mathbf{x}_0} p_{0t}(\mathbf{x}_0|\mathbf{x})\exp(-\beta\mathcal{E}(\mathbf{x}_0))\mathrm{d}\mathbf{x}_0\mathrm{d}\mathbf{x} \\
&= \int_{\mathbf{x}_0}\int_{\mathbf{x}} p_0(\mathbf{x}_0)p_{t0}(\mathbf{x}_t|\mathbf{x}_0)\exp(-\beta\mathcal{E}(\mathbf{x}_0))\mathrm{d}\mathbf{x}_0\mathrm{d}\mathbf{x} \\
&= \int_{\mathbf{x}_0} p_0(\mathbf{x}_0)\exp(-\beta\mathcal{E}(\mathbf{x}_0))\mathrm{d}\mathbf{x}_0 \\
&= \mathbb{E}_{\mathbf{x}_0}[\exp(-\beta\mathcal{E}(\mathbf{x}_0))], \tag{B.2}
\end{aligned}$$

where the second equation is due to the fact that $p_t(\mathbf{x})p_{0t}(\mathbf{x}_0|\mathbf{x}) = p(\mathbf{x},\mathbf{x}_0) = p_0(\mathbf{x}_0)p_{0t}(\mathbf{x}_0|\mathbf{x})$; and the third equation is due to the fact that $\int_{\mathbf{x}} p_{t0}(\mathbf{x}|\mathbf{x}_0)\mathrm{d}\mathbf{x} = 1$. Using B.2 to normalize $q_t(\mathbf{x}) \propto p_t(\mathbf{x})\exp(-\mathcal{E}_t(\mathbf{x}))$ yields the closed-form solution presented in (B.1). $\square$

Then we are ready to proof Theorem 4.1.

*Proof of Theorem 4.1.* We start by considering the condition flow $\mathbf{u}_{t0}(\mathbf{x}|\mathbf{x}_0)$ that generates the conditional distribution sequence $p_{t0}(\mathbf{x}|\mathbf{x}_0)$. According to the continuity equation (3.1), we have

$$\frac{\mathrm{d}p_{t0}(\mathbf{x}|\mathbf{x}_0)}{\mathrm{d}t} = -\mathrm{div}\cdot[p_{t0}(\mathbf{x}|\mathbf{x}_0)\mathbf{u}_{t0}(\mathbf{x}|\mathbf{x}_0)]. \tag{B.3}$$

Considering the dynamic of probability distribution $q_t(\mathbf{x})$, Lemma B.1 yields

$$
\begin{aligned}
\frac{\mathrm{d}q_t(\mathbf{x})}{\mathrm{d}t} &= \frac{\mathrm{d}}{\mathrm{d}t} \frac{p_t(\mathbf{x})\exp(-\mathcal{E}_t(\mathbf{x}))}{\mathbb{E}_{\mathbf{x}_0}[\exp(-\beta\mathcal{E}(\mathbf{x}_0))]} \\
&= \frac{\mathrm{d}}{\mathrm{d}t} \frac{p_t(\mathbf{x})\int_{\mathbf{x}_0} p_{0t}(\mathbf{x}_0|\mathbf{x})\exp(-\beta\mathcal{E}(\mathbf{x}_0))\mathrm{d}\mathbf{x}_0}{\mathbb{E}_{\mathbf{x}_0}[\exp(-\beta\mathcal{E}(\mathbf{x}_0))]} \\
&= \frac{\mathrm{d}}{\mathrm{d}t} \frac{\int_{\mathbf{x}_0} p(\mathbf{x}_0,\mathbf{x})\exp(-\beta\mathcal{E}(\mathbf{x}_0))\mathrm{d}\mathbf{x}_0}{\mathbb{E}_{\mathbf{x}_0}[\exp(-\beta\mathcal{E}(\mathbf{x}_0))]} \\
&= \frac{\mathrm{d}}{\mathrm{d}t} \frac{\int_{\mathbf{x}_0} p_{t0}(\mathbf{x}|\mathbf{x}_0)p_0(\mathbf{x}_0)\exp(-\beta\mathcal{E}(\mathbf{x}_0))\mathrm{d}\mathbf{x}_0}{\mathbb{E}_{\mathbf{x}_0}[\exp(-\beta\mathcal{E}(\mathbf{x}_0))]} \\
&= \frac{1}{\mathbb{E}_{\mathbf{x}_0}[\exp(-\beta\mathcal{E}(\mathbf{x}_0))]} \int_{\mathbf{x}_0} \frac{\mathrm{d}}{\mathrm{d}t} p_{t0}(\mathbf{x}|\mathbf{x}_0)p_0(\mathbf{x}_0)\exp(-\beta\mathcal{E}(\mathbf{x}_0))\mathrm{d}\mathbf{x}_0, \quad\text{(B.4)}
\end{aligned}
$$

where the second equation is due to the definition of $\mathcal{E}_t(\mathbf{x})$; the third and the forth equation is due to the Bayesian equality $p_t(\mathbf{x})p_{0t}(\mathbf{x}_0|\mathbf{x}) = p(\mathbf{x}_0,\mathbf{x}) = p_0(\mathbf{x}_0)p_{t0}(\mathbf{x}|\mathbf{x}_0)$; and the last equation is due to the fact that the denominator $\mathbb{E}_{\mathbf{x}_0}[-\exp(\beta\mathcal{E}(\mathbf{x}_0))]$ is a constant with respect to time $t$. Plugging the definition of continuity equation (3.1) into (B.4) suggests that

$$
\begin{aligned}
\frac{\mathrm{d}q_t(\mathbf{x})}{\mathrm{d}t} &= \frac{1}{\mathbb{E}_{\mathbf{x}_0}[\exp(-\beta\mathcal{E}(\mathbf{x}_0))]} \int_{\mathbf{x}_0} \frac{\mathrm{d}}{\mathrm{d}t} p_{t0}(\mathbf{x}|\mathbf{x}_0)p_0(\mathbf{x}_0)\exp(-\beta\mathcal{E}(\mathbf{x}_0))\mathrm{d}\mathbf{x}_0 \\
&= -\frac{\int_{\mathbf{x}_0} \mathrm{div}\cdot[p_{t0}(\mathbf{x}|\mathbf{x}_0)\mathbf{u}_{t0}(\mathbf{x}|\mathbf{x}_0)]\,p_0(\mathbf{x}_0)\exp(-\beta\mathcal{E}(\mathbf{x}_0))\mathrm{d}\mathbf{x}_0}{\mathbb{E}_{\mathbf{x}_0}[\exp(-\beta\mathcal{E}(\mathbf{x}_0))]} \\
&= -\frac{\mathrm{div}\cdot\left[\int_{\mathbf{x}_0} p_{t0}(\mathbf{x}|\mathbf{x}_0)\mathbf{u}_{t0}(\mathbf{x}|\mathbf{x}_0)p_0(\mathbf{x}_0)\exp(-\beta\mathcal{E}(\mathbf{x}_0))\mathrm{d}\mathbf{x}_0\right]}{\mathbb{E}_{\mathbf{x}_0}[\exp(-\beta\mathcal{E}(\mathbf{x}_0))]} \\
&= -\mathrm{div}\cdot\left[\frac{p_t(\mathbf{x})\exp(-\mathcal{E}_t(\mathbf{x}))}{\mathbb{E}_{\mathbf{x}_0}[\exp(-\beta\mathcal{E}(\mathbf{x}_0))]}\int_{\mathbf{x}_0} p_{0t}(\mathbf{x}_0|\mathbf{x})\mathbf{u}_{t0}(\mathbf{x}|\mathbf{x}_0)\frac{\exp(-\beta\mathcal{E}(\mathbf{x}_0))}{\exp(-\mathcal{E}_t(\mathbf{x}))}\mathrm{d}\mathbf{x}_0\right] \quad\text{(B.5)} \\
&= -\mathrm{div}\cdot[q_t(\mathbf{x})\widehat{\mathbf{u}}_t(\mathbf{x})],
\end{aligned}
$$

where the third and the fourth equation is due to the linearity of the divergence operator $\mathrm{div}\cdot[\cdot]$. The fifth equality is due to the definition of $q_t(\mathbf{x})$ and $\widehat{\mathbf{u}}_t(\mathbf{x})$. Thus we conclude our proof by showing $\widehat{\mathbf{u}}_t(\mathbf{x})$ defined by (B.5) can generate the guided distribution sequence $q_t(\mathbf{x})$. $\qquad\square$

## B.2 PROOF OF THEOREM 4.3

*Proof of Theorem 4.3.* We start by fixing time $t$ and writing the gradient of $\|\mathbf{v}_t^{\boldsymbol{\theta}}(\mathbf{x}) - \widehat{\mathbf{u}}_t(\mathbf{x})\|_2^2$ with respect to $\boldsymbol{\theta}$ as

$$
\begin{aligned}
\nabla_{\boldsymbol{\theta}}\|\mathbf{v}_t^{\boldsymbol{\theta}}(\mathbf{x}) - \widehat{\mathbf{u}}_t(\mathbf{x})\|_2^2 &= \nabla_{\boldsymbol{\theta}}\left\|\mathbf{v}_t^{\boldsymbol{\theta}}(\mathbf{x})\right\|_2^2 - 2\left\langle\nabla_{\boldsymbol{\theta}}\mathbf{v}_t^{\boldsymbol{\theta}}(\mathbf{x}),\widehat{\mathbf{u}}_t(\mathbf{x})\right\rangle + \nabla_{\boldsymbol{\theta}}\|\widehat{\mathbf{u}}_t(\mathbf{x})\|_2^2 \\
&= \nabla_{\boldsymbol{\theta}}\left\|\mathbf{v}_t^{\boldsymbol{\theta}}(\mathbf{x})\right\|_2^2 - 2\left\langle\nabla_{\boldsymbol{\theta}}\mathbf{v}_t^{\boldsymbol{\theta}}(\mathbf{x}),\widehat{\mathbf{u}}_t(\mathbf{x})\right\rangle, \quad\text{(B.6)}
\end{aligned}
$$

where we drop the third term in the last equation due to the fact that $\nabla_{\boldsymbol{\theta}}\|\widehat{\mathbf{u}}_t(\mathbf{x})\|_2^2 = 0$. Consider the gradient of loss $\mathcal{L}_{\mathrm{CEFM}}(\boldsymbol{\theta})$, we have

$$
\begin{aligned}
&\nabla_{\boldsymbol{\theta}}\mathcal{L}_{\mathrm{CEFM}}(\boldsymbol{\theta}) \\
&= \nabla_{\boldsymbol{\theta}}\iint_{\mathbf{x},\mathbf{x}_0} p_0(\mathbf{x}_0)p_{t0}(\mathbf{x}|\mathbf{x}_0)\frac{\exp(-\beta\mathcal{E}(\mathbf{x}_0))}{\mathbb{E}_{p_0(\mathbf{x}_0)}[\exp(-\beta\mathcal{E}(\mathbf{x}_0))]}\left\|\mathbf{v}_t^{\boldsymbol{\theta}}(\mathbf{x}) - \mathbf{u}_{t0}(\mathbf{x}|\mathbf{x}_0)\right\|_2^2\mathrm{d}\mathbf{x}\mathrm{d}\mathbf{x}_0 \\
&= \iint_{\mathbf{x},\mathbf{x}_0} p_t(\mathbf{x})p_{0t}(\mathbf{x}_0|\mathbf{x})\frac{\exp(-\beta\mathcal{E}(\mathbf{x}_0))}{\mathbb{E}_{p_0(\mathbf{x}_0)}[\exp(-\beta\mathcal{E}(\mathbf{x}_0))]}\nabla_{\boldsymbol{\theta}}\left\|\mathbf{v}_t^{\boldsymbol{\theta}}(\mathbf{x})\right\|_2^2\mathrm{d}\mathbf{x}\mathrm{d}\mathbf{x}_0 \\
&\quad - 2\iint_{\mathbf{x},\mathbf{x}_0} p_t(\mathbf{x})p_{0t}(\mathbf{x}_0|\mathbf{x})\frac{\exp(-\beta\mathcal{E}(\mathbf{x}_0))}{\mathbb{E}_{p_0(\mathbf{x}_0)}[\exp(-\beta\mathcal{E}(\mathbf{x}_0))]}\left\langle\nabla_{\boldsymbol{\theta}}\mathbf{v}_t^{\boldsymbol{\theta}}(\mathbf{x}),\mathbf{u}_{t0}(\mathbf{x}|\mathbf{x}_0)\right\rangle\mathrm{d}\mathbf{x}\mathrm{d}\mathbf{x}_0, \quad\text{(B.7)}
\end{aligned}
$$

where the second equation leverages the Bayesian rule $p_t(\mathbf{x})p_{0t}(\mathbf{x}|\mathbf{x}_0) = p(\mathbf{x}, \mathbf{x}_0) = p_0(\mathbf{x}_0)p_{t0}(\mathbf{x}|\mathbf{x}_0)$ and the fact that $\nabla_{\boldsymbol{\theta}}\|\mathbf{u}_{t0}(\mathbf{x}|\mathbf{x}_0)\|_2^2 = 0$. Noticing the fact that

$$\exp(-\mathcal{E}_t(\mathbf{x})) = \mathbb{E}_{p_{0t}(\mathbf{x}_0|\mathbf{x}_t)}[\exp(-\beta\mathcal{E}(\mathbf{x}_0))] = \int_{\mathbf{x}_0} p_{0t}(\mathbf{x}_0|\mathbf{x})\exp(-\beta\mathcal{E}(\mathbf{x}_0))\mathrm{d}\mathbf{x}_0, \quad \text{(B.8)}$$

$$\mathbb{E}_{p_t(\mathbf{x})}[\exp(-\mathcal{E}_t(\mathbf{x}))] = \iint_{\mathbf{x},\mathbf{x}_0} p_t(\mathbf{x})p_{0t}(\mathbf{x}_0|\mathbf{x})\exp(-\beta\mathcal{E}(\mathbf{x}_0))\mathrm{d}\mathbf{x}_0\mathrm{d}\mathbf{x} = \mathbb{E}_{p_0(\mathbf{x}_0)}[\exp(-\beta\mathcal{E}(\mathbf{x}_0))]. \tag{B.9}$$

By reorganizing the integral, (B.7) yields

$$
\begin{aligned}
&\nabla_{\boldsymbol{\theta}}\mathcal{L}_{\text{CEFM}}(\boldsymbol{\theta})\\
&= \int_{\mathbf{x}} p_t(\mathbf{x})\frac{\exp(-\mathcal{E}_t(\mathbf{x}))}{\mathbb{E}_{p_t(\mathbf{x})}[\exp(-\mathcal{E}_t(\mathbf{x}))]}\nabla_{\boldsymbol{\theta}}\left\|\mathbf{v}_t^{\boldsymbol{\theta}}(\mathbf{x})\right\|_2^2\mathrm{d}\mathbf{x}\\
&\quad - 2\int_{\mathbf{x}}\frac{p_t(\mathbf{x})\exp(-\mathcal{E}_t(\mathbf{x}))}{\mathbb{E}_{p_t(\mathbf{x})}[\exp(-\mathcal{E}_t(\mathbf{x}))]}\left\langle\nabla_{\boldsymbol{\theta}}\mathbf{v}_t^{\boldsymbol{\theta}}(\mathbf{x}), \int_{\mathbf{x}_0} p_{0t}(\mathbf{x}_0|\mathbf{x})\frac{\exp(-\beta\mathcal{E}(\mathbf{x}_0))}{\exp(-\mathcal{E}_t(\mathbf{x}))}\mathbf{u}_{t0}(\mathbf{x}|\mathbf{x}_0)\mathrm{d}\mathbf{x}_0\right\rangle\mathrm{d}\mathbf{x}\\
&= \int_{\mathbf{x}} p_t(\mathbf{x})\frac{\exp(-\mathcal{E}_t(\mathbf{x}))}{\mathbb{E}_{p_t(\mathbf{x})}[\exp(-\mathcal{E}_t(\mathbf{x}))]}\left(\nabla_{\boldsymbol{\theta}}\|\mathbf{v}_t^{\boldsymbol{\theta}}(\mathbf{x})\|_2^2 - 2\langle\nabla_{\boldsymbol{\theta}}\mathbf{v}_t^{\boldsymbol{\theta}}(\mathbf{x}), \widehat{\mathbf{u}}_t(\mathbf{x})\rangle\right)\mathrm{d}\mathbf{x}\\
&= \int_{\mathbf{x}} p_t(\mathbf{x})\frac{\exp(-\mathcal{E}_t(\mathbf{x}))}{\mathbb{E}_{p_t(\mathbf{x})}[\exp(-\mathcal{E}_t(\mathbf{x}))]}\nabla_{\boldsymbol{\theta}}\|\mathbf{v}_t^{\boldsymbol{\theta}}(\mathbf{x}) - \widehat{\mathbf{u}}_t(\mathbf{x})\|_2^2\mathrm{d}\mathbf{x}\\
&= \nabla_{\boldsymbol{\theta}}\mathcal{L}_{\text{EFM}}(\boldsymbol{\theta}),
\end{aligned}
\tag{B.10}
$$

where the second equation is the combination of (B.9) and the definition of $\widehat{\mathbf{u}}_t(\mathbf{x})$ declared in (4.1) in Theorem 4.1. Thus we complete the proof. $\qquad\square$

## B.3  PROOF OF COROLLARY 4.7

*Proof of Corollary 4.7.* We start by denoting $f_t = \dot{\mu}_t/\mu_t$ and $g_t^2 = (\dot{\mu}_t\sigma_t - \mu_t\dot{\sigma}_t)\sigma_t/\mu_t$. According to Lemma 3.2, we have

$$\widehat{\mathbf{u}}_t(\mathbf{x}) = f_t\mathbf{x} + g_t^2\nabla_{\mathbf{x}}\log q_t(\mathbf{x}), \quad \mathbf{u}_{t0}(\mathbf{x}|\mathbf{x}_0) = f_t\mathbf{x} + g_t^2\nabla_{\mathbf{x}}\log p_{t0}(\mathbf{x}|\mathbf{x}_0), \tag{B.11}$$

then for the RHS in (4.4),

$$
\begin{aligned}
&\int_{\mathbf{x}_0} p_{0t}(\mathbf{x}_0|\mathbf{x})\mathbf{u}_{t0}(\mathbf{x}|\mathbf{x}_0)\frac{\exp(-\beta\mathcal{E}(\mathbf{x}_0))}{\exp(-\mathcal{E}_t(\mathbf{x}))}\mathrm{d}\mathbf{x}_0\\
&= \int_{\mathbf{x}_0} p_{0t}(\mathbf{x}_0|\mathbf{x})\left(f_t\mathbf{x} + g_t^2\nabla_{\mathbf{x}}\log p_{t0}(\mathbf{x}|\mathbf{x}_0)\right)\frac{\exp(-\beta\mathcal{E}(\mathbf{x}_0))}{\exp(-\mathcal{E}_t(\mathbf{x}))}\mathrm{d}\mathbf{x}_0\\
&= f_t\mathbf{x}\int_{\mathbf{x}_0} p_{0t}(\mathbf{x}_0|\mathbf{x})\frac{\exp(-\beta\mathcal{E}(\mathbf{x}_0))}{\mathbb{E}_{p_{0t}}[\exp(-\beta\mathcal{E}(\mathbf{x}_0))]}\mathrm{d}\mathbf{x}_0\\
&\quad + g_t^2\int_{\mathbf{x}_0} p_{0t}(\mathbf{x}_0|\mathbf{x})\nabla_{\mathbf{x}}\log p_{t0}(\mathbf{x}|\mathbf{x}_0)\frac{\exp(-\beta\mathcal{E}(\mathbf{x}_0))}{\exp(-\mathcal{E}_t(\mathbf{x}))}\mathrm{d}\mathbf{x}_0\\
&= f_t\mathbf{x} + g_t^2\int_{\mathbf{x}_0} p_{0t}(\mathbf{x}_0|\mathbf{x})\nabla_{\mathbf{x}}\log p_{t0}(\mathbf{x}|\mathbf{x}_0)\frac{\exp(-\beta\mathcal{E}(\mathbf{x}_0))}{\exp(-\mathcal{E}_t(\mathbf{x}))}\mathrm{d}\mathbf{x}_0,
\end{aligned}
\tag{B.12}
$$

where the first equation is due to equation (B.11). The second equation is due to the linearity of the integral and the third equation utilizes the fact that $\int_{\mathbf{x}_0} p_{0t}(\mathbf{x}_0|\mathbf{x})\exp(-\beta\mathcal{E}(\mathbf{x}_0))\mathrm{d}\mathbf{x}_0 = \mathbb{E}_{p_{0t}(\mathbf{x}_0|\mathbf{x})}[\exp(-\beta\mathcal{E}(\mathbf{x}_0))] = \exp(-\mathcal{E}_t(\mathbf{x}))$. Plugging (B.12) back into Theorem 4.1 and utilizing (B.11) yields

$$f_t\mathbf{x} + g_t^2\nabla_{\mathbf{x}}\log q_t(\mathbf{x}) = \widehat{\mathbf{u}}(\mathbf{x}) = f_t\mathbf{x} + g_t^2\int_{\mathbf{x}_0} p_{0t}(\mathbf{x}_0|\mathbf{x})\nabla_{\mathbf{x}}\log p_{t0}(\mathbf{x}|\mathbf{x}_0)\frac{\exp(-\beta\mathcal{E}(\mathbf{x}_0))}{\exp(-\mathcal{E}_t(\mathbf{x}))}\mathrm{d}\mathbf{x}_0,$$

which completes by canceling the $f_t\mathbf{x}$ term on both side of the equation. $\qquad\square$

## B.4 PROOF OF COROLLARY 4.8

*Proof.* The proof is similar with the proof of Theorem 4.3 by replacing $\mathbf{v}_t^{\boldsymbol{\theta}}$ with $\mathbf{s}_t^{\boldsymbol{\theta}}$, replacing $\mathbf{u}_{t0}(\mathbf{x}|\mathbf{x}_0)$ with $\nabla_{\mathbf{x}} \log p_{t0}(\mathbf{x}|\mathbf{x}_0)$. Following the same rule in (B.7) and (B.10) yields

$$
\nabla_{\boldsymbol{\theta}} \mathcal{L}_{\mathrm{CED}}(\boldsymbol{\theta})
$$
$$
= \nabla_{\boldsymbol{\theta}} \iint_{\mathbf{x},\mathbf{x}_0} p_0(\mathbf{x}_0) p_{t0}(\mathbf{x}|\mathbf{x}_0) \frac{\exp(-\beta \mathcal{E}(\mathbf{x}_0))}{\mathbb{E}_{p_0(\mathbf{x}_0)}[\exp(-\beta \mathcal{E}(\mathbf{x}_0))]} \left\| \mathbf{s}_t^{\boldsymbol{\theta}}(\mathbf{x}) - \nabla_{\mathbf{x}} \log p_{t0}(\mathbf{x}|\mathbf{x}_0) \right\|_2^2 \mathrm{d}\mathbf{x}\mathrm{d}\mathbf{x}_0
$$
$$
= \iint_{\mathbf{x},\mathbf{x}_0} p_t(\mathbf{x}) p_{0t}(\mathbf{x}_0|\mathbf{x}) \frac{\exp(-\beta \mathcal{E}(\mathbf{x}_0))}{\mathbb{E}_{p_0(\mathbf{x}_0)}[\exp(-\beta \mathcal{E}(\mathbf{x}_0))]} \nabla_{\boldsymbol{\theta}} \left\| \mathbf{s}_t^{\boldsymbol{\theta}}(\mathbf{x}) \right\|_2^2 \mathrm{d}\mathbf{x}\mathrm{d}\mathbf{x}_0
$$
$$
- 2 \iint_{\mathbf{x},\mathbf{x}_0} p_t(\mathbf{x}) p_{0t}(\mathbf{x}_0|\mathbf{x}) \frac{\exp(-\beta \mathcal{E}(\mathbf{x}_0))}{\mathbb{E}_{p_0(\mathbf{x}_0)}[\exp(-\beta \mathcal{E}(\mathbf{x}_0))]} \left\langle \nabla_{\boldsymbol{\theta}} \mathbf{s}_t^{\boldsymbol{\theta}}(\mathbf{x}), \nabla_{\mathbf{x}} \log p_{t0}(\mathbf{x}|\mathbf{x}_0) \right\rangle \mathrm{d}\mathbf{x}\mathrm{d}\mathbf{x}_0
$$
$$
= \int_{\mathbf{x}} p_t(\mathbf{x}) \frac{\exp(-\mathcal{E}_t(\mathbf{x}))}{\mathbb{E}_{p_t(\mathbf{x})}[\exp(-\mathcal{E}_t(\mathbf{x}))]} \nabla_{\boldsymbol{\theta}} \left\| \mathbf{s}_t^{\boldsymbol{\theta}}(\mathbf{x}) \right\|_2^2 \mathrm{d}\mathbf{x}
$$
$$
- 2 \int_{\mathbf{x}} \frac{p_t(\mathbf{x}) \exp(-\mathcal{E}_t(\mathbf{x}))}{\mathbb{E}_{p_t(\mathbf{x})}[\exp(-\mathcal{E}_t(\mathbf{x}))]} \left\langle \nabla_{\boldsymbol{\theta}} \mathbf{s}_t^{\boldsymbol{\theta}}(\mathbf{x}), \int_{\mathbf{x}_0} p_{0t}(\mathbf{x}_0|\mathbf{x}) \frac{\exp(-\beta \mathcal{E}(\mathbf{x}_0))}{\exp(-\mathcal{E}_t(\mathbf{x}))} \nabla_{\mathbf{x}} \log p_{t0}(\mathbf{x}|\mathbf{x}_0) \mathrm{d}\mathbf{x}_0 \right\rangle \mathrm{d}\mathbf{x}
$$
$$
= \int_{\mathbf{x}} p_t(\mathbf{x}) \frac{\exp(-\mathcal{E}_t(\mathbf{x}))}{\mathbb{E}_{p_t(\mathbf{x})}[\exp(-\mathcal{E}_t(\mathbf{x}))]} \left( \nabla_{\boldsymbol{\theta}} \| \mathbf{v}_t^{\boldsymbol{\theta}}(\mathbf{x})\|_2^2 - 2\langle \nabla_{\boldsymbol{\theta}} \mathbf{s}_t^{\boldsymbol{\theta}}(\mathbf{x}), \nabla_{\mathbf{x}} \log q_t(\mathbf{x}) \rangle \right) \mathrm{d}\mathbf{x}
$$
$$
= \int_{\mathbf{x}} p_t(\mathbf{x}) \frac{\exp(-\mathcal{E}_t(\mathbf{x}))}{\mathbb{E}_{p_t(\mathbf{x})}[\exp(-\mathcal{E}_t(\mathbf{x}))]} \nabla_{\boldsymbol{\theta}} \| \mathbf{s}_t^{\boldsymbol{\theta}}(\mathbf{x}) - \nabla_{\mathbf{x}} \log q_t(\mathbf{x})\|_2^2 \mathrm{d}\mathbf{x}
$$
$$
= \nabla_{\boldsymbol{\theta}} \mathcal{L}_{\mathrm{ED}}(\boldsymbol{\theta})
$$

$\square$

## B.5 PROOF OF LEMMA 4.10

We present the proof for the classifier guidance (CG) and classifier-free guidance (CFG) presented in (4.6) in this subsection. The score function for Contrastive Energy Prediction (CEP) and Energy-weighted Diffusion (ED) presented in (4.7) has been extensively discussed in Corollary 4.7.

*Proof of* (4.6) *in Lemma 4.10.* In order to sample from the guided distribution $q$, CG and CFG estimate the score function $\nabla_{\mathbf{x}} q_t(\mathbf{x})$ as

$$
\nabla_{\mathbf{x}} \log q_t(\mathbf{x}) = \nabla_{\mathbf{x}} \log p_t(\mathbf{x}) + \beta \nabla_{\mathbf{x}} \log p_t(c|\mathbf{x}) = (1 - \beta) \nabla_{\mathbf{x}} \log p_t(\mathbf{x}) + \beta \nabla_{\mathbf{x}} \log p_t(\mathbf{x}|c), \tag{B.13}
$$

where the second equation is given by the classifier free guidance. Applying the Bayesian rule, $p_t(c|\mathbf{x})$ can be decomposed by

$$
p_t(c|\mathbf{x}) = \int_{\mathbf{x}_1} p(\mathbf{x}_1|\mathbf{x}) p(c|\mathbf{x}_1) \mathrm{d}\mathbf{x}_1 = \mathbb{E}_{p_{1t}(\mathbf{x}_1|\mathbf{x})} p(c|\mathbf{x}_1). \tag{B.14}
$$

Plugging (B.14) into (B.13), the score function $\nabla_{\mathbf{x}} \log q_t(\mathbf{x})$ is written by

$$
\nabla_{\mathbf{x}} \log q_t(\mathbf{x}) = \nabla_{\mathbf{x}} \log p_t(\mathbf{x}) + \beta \nabla_{\mathbf{x}} \log \mathbb{E}_{p_{1t}(\mathbf{x}_1|\mathbf{x})} p(c|\mathbf{x}_1)
$$
$$
= \nabla_{\mathbf{x}} \log p_t(\mathbf{x}) + \nabla_{\mathbf{x}} \left[ \log \mathbb{E}_{p_{1t}(\mathbf{x}_1|\mathbf{x})} p(c|\mathbf{x}_1) \right]^{\beta}
$$

where the last equation absorbs the $\beta$ into logarithmic operator, which concludes the proof. $\square$

## B.6 Proof of Lemma 4.11

*Proof.* Plugging $\mathcal{E}(\mathbf{x}_0)) = -\log p(c|\mathbf{x}_0)$ and $\beta = 1$ into the definition of $\mathcal{L}_{\text{CED}}$ suggests that

$$
\begin{aligned}
\mathcal{L}_{\text{CED}}(\boldsymbol{\theta}) &= \mathbb{E}_{t,\mathbf{x},\mathbf{x}_0}\left[\frac{p(c|\mathbf{x}_0)}{\mathbb{E}_{p_0(\mathbf{x}_0)}[p(c|\mathbf{x}_0)]}\left\|\mathbf{s}_t^{\boldsymbol{\theta}}(\mathbf{x}) - \nabla_{\mathbf{x}}\log p_{t0}(\mathbf{x}|\mathbf{x}_0)\right\|_2^2\right] \\
&= \iiint_{t,\mathbf{x},\mathbf{x}_0}\frac{\lambda(t)p(c|\mathbf{x}_0)p_0(\mathbf{x}_0)p_{t0}(\mathbf{x}|\mathbf{x}_0)}{p(c)}\left\|\mathbf{s}_t^{\boldsymbol{\theta}}(\mathbf{x}) - \nabla_{\mathbf{x}}\log p_{t0}(\mathbf{x}|\mathbf{x}_0)\right\|_2^2 dt\,d\mathbf{x}\,d\mathbf{x}_0 \\
&= \iiint_{t,\mathbf{x},\mathbf{x}_0}\lambda(t)p(\mathbf{x}_0|c)p_{t0}(\mathbf{x}|\mathbf{x}_0)\left\|\mathbf{s}_t^{\boldsymbol{\theta}}(\mathbf{x}) - \nabla_{\mathbf{x}}\log p_{t0}(\mathbf{x}|\mathbf{x}_0)\right\|_2^2 dt\,d\mathbf{x}\,d\mathbf{x}_0 \\
&= \mathbb{E}_{t,\mathbf{x}_0,\mathbf{x}|c}\left[\left\|\mathbf{s}_t^{\boldsymbol{\theta}}(\mathbf{x}) - \nabla_{\mathbf{x}}\log p_{t0}(\mathbf{x}|\mathbf{x}_0)\right\|_2^2\right], \quad\quad\quad (B.15)
\end{aligned}
$$

where the second equation expands the expectation and write $\mathbb{E}_{p_0(\mathbf{x})}[p(c|\mathbf{x}_0)] = p(c)$ due to Bayesian rule. The third equation is due to $p(c|\mathbf{x}_0)p_0(\mathbf{x}_0)/p(c) = p(\mathbf{x}_0|c)$. The fourth equation rewrites the integral back as expectation. The optimal solution for (B.15) is $\mathbf{s}_t^{\boldsymbol{\theta}}(\mathbf{x}) = \nabla_{\mathbf{x}}\log p_t(\mathbf{x}|c)$, given the analysis for the standard score-matching process (Song et al., 2020). $\square$

## C Comparison between Classifier Guidance and Energy Guidance

We visualize the differences between the classifier-guidance and energy guidance in this section.

**Experiment setup.** We follow the setting in Lu et al. (2023) to sample the data distribution $p(\mathbf{x})$ and energy function $\mathcal{E}(\mathbf{x})$. We shift the energy function by $\mathcal{E}(\mathbf{x}) \leftarrow \mathcal{E}(\mathbf{x}) - \min_{\mathbf{x}}\mathcal{E}(\mathbf{x})$ so that all energy $\mathcal{E}(\mathbf{x}) \geq 0$. We note that this translation does not change the distribution of $q(\mathbf{x}) \propto p(\mathbf{x})\exp(-\beta\mathcal{E}(\mathbf{x}))$. According to $\mathcal{E}(\mathbf{x}) = -\log p(c|\mathbf{x})$ discussed in Section 4.3, we generate the binary class $c$ such that $p(c = 1|\mathbf{x}) = \exp(-\mathcal{E}(\mathbf{x}))$. The ground truth distribution of $q(\mathbf{x}) \propto p(\mathbf{x})\exp(-\beta\mathcal{E}(\mathbf{x})) = p(\mathbf{x})p^\beta(c = 1|\mathbf{x})$ with different $\beta$, the posterior distribution $p(\mathbf{x}|c = 0)$, $p(\mathbf{x}|c = 1)$ is presented in Figure 2.

From Figure 2 we can find that when $\beta = 1$, $p(\mathbf{x})p(c = 1|\mathbf{x}) \propto p(\mathbf{x}|c = 1)$. However, when $\beta \geq 1$, $p(\mathbf{x})\exp(-\beta\mathcal{E}(\mathbf{x})) = p(\mathbf{x})p^\beta(c = 1|\mathbf{x})$ will be more concentrated on the region with higher likelihood $p(c = 1|\mathbf{x})$. We train the score function by optimizing $\mathcal{L}_{\text{CED}}$ as of Algorithm 1, and the result is presented in Figure 3a. In parallel to the implementation of our methods, we train the score function $\mathbf{s}_t^{\boldsymbol{\theta}}(\mathbf{x}, \emptyset)$ and $\mathbf{s}_t^{\boldsymbol{\theta}}(\mathbf{x}, c = 1)$, and implement the classifier-free guidance by

$$\nabla_{\boldsymbol{\theta}}\log q_t(\mathbf{x}) = (1 - \beta)\mathbf{s}_t^{\boldsymbol{\theta}}(\mathbf{x}, \emptyset) + \beta\mathbf{s}_t^{\boldsymbol{\theta}}(\mathbf{x}, c = 1) \quad\quad\quad (C.1)$$

as presented in Figure 3b.

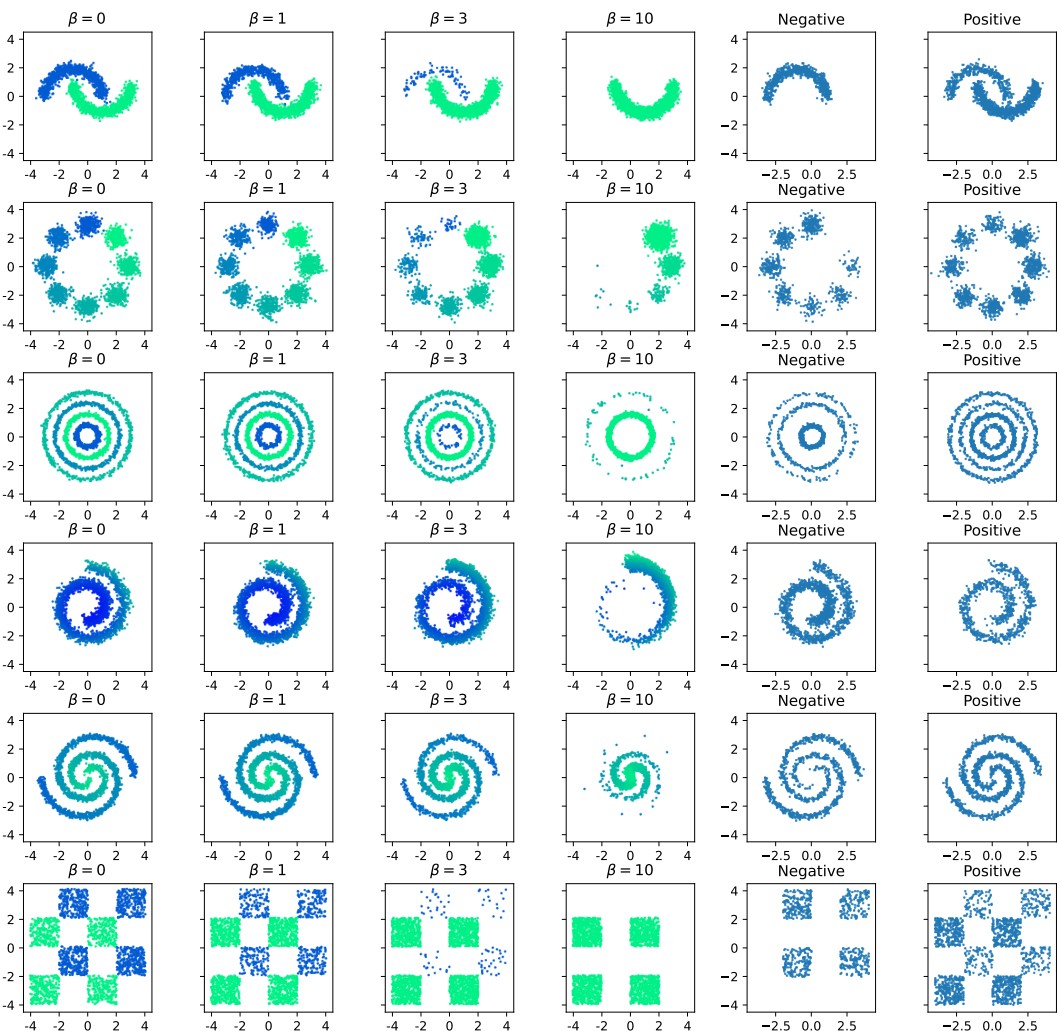

Figure 2: The ground truth distribution of $q(\mathbf{x}) \propto p(\mathbf{x})\exp(-\beta\mathcal{E}(\mathbf{x})) = p(\mathbf{x})p^\beta(c = 1|\mathbf{x})$ with different $\beta$, the posterior distribution $p(\mathbf{x}|c = 0)$ (Negative) and $p(\mathbf{x}|c = 1)$ (Positive) over 6 data distributions.

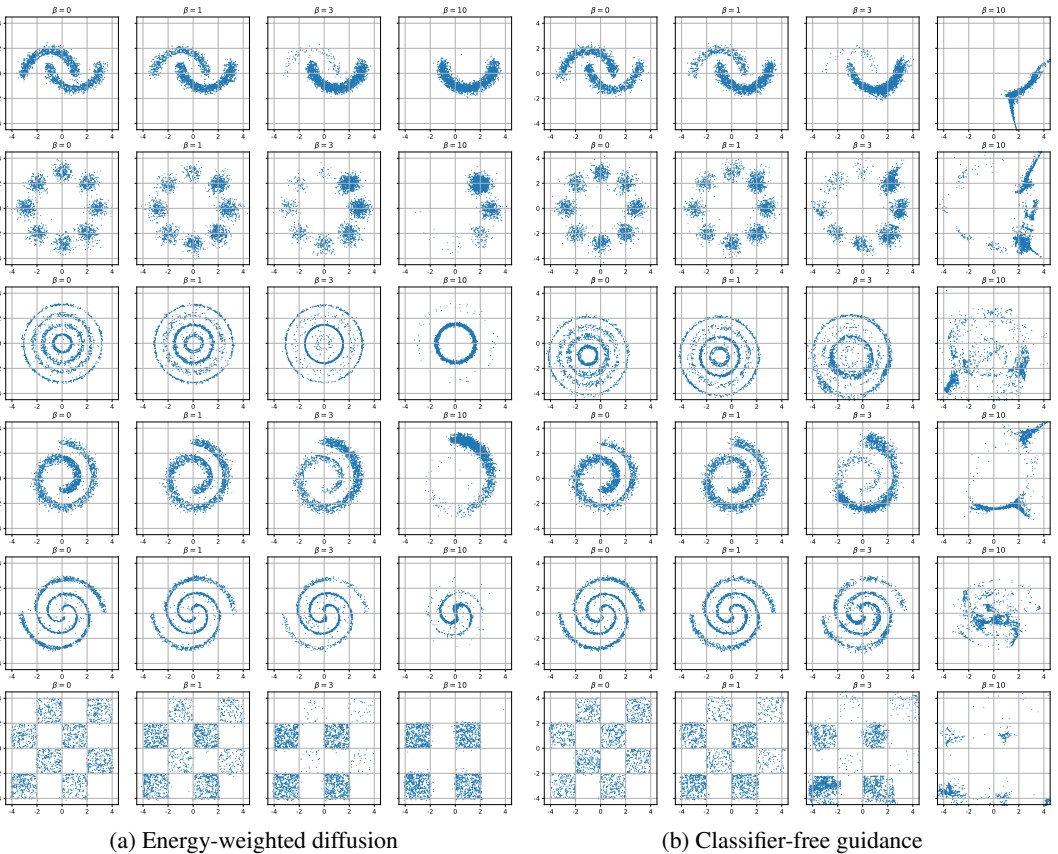

(a) Energy-weighted diffusion          (b) Classifier-free guidance

Figure 3: The data sampled from the score function trained by energy-weighted diffusion and the score function composed by classifier-free guidance with different guidance scale $\beta$. We sample 2000 data points for each state.

# D  DETAILED ALGORITHMS

We present the detail of the algorithms that are skipped in Section 4 and Section 5. We present the energy-weighted flow matching model in Algorithms 3.

---

**Algorithm 3** Training Energy-Weighted Flow Matching Model

---

**Input:** Velocity vector field $\mathbf{v}_t^{\boldsymbol{\theta}}(\cdot)$, guidance scale $\beta$, batch size $B$, time weight $\lambda(t)$
**Input:** Conditional vector field $\mathbf{u}_{t0}(\mathbf{x}|\mathbf{x}_0)$, conditional distribution $p_{t0}$ generated by $\mathbf{u}_{t0}$.
 1: **for** batch $\{\mathbf{x}_0^i, \mathcal{E}(\mathbf{x}_0^i)\}_i$ **do**
 2:     **for** index $i \in [B]$ **do**
 3:         Sample $t_i \sim U(0,1)$, sample $\mathbf{x}_i \sim p_{t_i 0}(\cdot|\mathbf{x}_0^i)$
 4:         Calculate guidance $g_i = \text{softmax}(-\beta \mathcal{E}(\mathbf{x}_0^i)) = \exp(-\beta \mathcal{E}(\mathbf{x}_0^i))/\sum_j \exp(-\beta \mathcal{E}(\mathbf{x}_0^j))$
 5:     **end for**
 6:     Calculate and take a gradient step using $\mathcal{L}_{\text{CEFM}}(\boldsymbol{\theta}) = \sum_i \lambda(t_i) g_i \|\mathbf{v}_{t_i}^{\boldsymbol{\theta}}(\mathbf{x}_{t_i}) - \mathbf{u}_{t_i 0}(\mathbf{x}_i|\mathbf{x}_0^i)\|_2^2$.
 7: **end for**

---

**Q-weighted Iterative Policy Optimization (QIPO).** We use the in-support softmax $Q$-learning (Lu et al., 2023) to implement the QIPO. We would like to emphasize that any $Q$ function given by the offline RL algorithm is compatible with our framework. The flow matching version is presented in Algorithm 4 and the diffusion version is presented in Algorithm 5. We use $Q_\perp^\psi$ to represent the $Q$ function not contributing gradient to the loss, which is commonly used in TD-learning.

---

**Algorithm 4** Q-weighted iterative policy optimization for offline RL (Flow Matching)

---

**Input:** Vector field function $\mathbf{v}_t^{\boldsymbol{\theta}}(\cdot)$, guidance scale $\beta$, batch size $B$, weight on time $\lambda(t)$
**Input:** Conditional vector field $\mathbf{u}_{t0}(\mathbf{x}|\mathbf{x}_0)$, conditional distribution $p_{t0}$ generated by $\mathbf{u}_{t0}$.
**Input:** Offline RL dataset $\mathcal{D} = \{(\mathbf{x}, \mathbf{a}, \mathbf{x}', r)\}$, number of epochs $K_1$, $K_2$ and $K_3$, $Q$ function $Q^\psi$
**Input:** Support action set size $M$, support action set renew frequency $K_{\text{renew}}$
 1: **for** diffusion model warm-up epoch $k \in [K_1]$ **do**
 2:     **for** batch $\{(\mathbf{x}_i, \mathbf{a}_i, \mathbf{x}_i', r_i)\}_i \subset \mathcal{D}$ **do**
 3:         Sample $t_i \sim U(0,1), \mathbf{a}_{i,t} \sim p_{t_i 0}(\cdot|\mathbf{a}_i)$ for all $i \in [B]$
 4:         Calculate and optimize $\mathcal{L}_{\text{QF, warmup}}(\boldsymbol{\theta}) = \sum_{i=1}^B \lambda(t_i) \|\mathbf{v}_t^{\boldsymbol{\theta}}(\mathbf{a}_{i,t}; \mathbf{x}_i) - \mathbf{u}_{t_i 0}(\mathbf{a}_{i,t}|\mathbf{a}_i)\|_2^2$
 5:     **end for**
 6: **end for**
 7: **for** Q-learning warm-up epoch $k \in [K_2]$ **do**
 8:     **for** batch $\{(\mathbf{x}_i, \mathbf{a}_i, \mathbf{x}_i', r_i)\}_i \subset \mathcal{D}$ **do**
 9:         Sample support action set $\mathbf{a}_{ij}'$ using vector field $\mathbf{v}_t^{\boldsymbol{\theta}}$ for $j \in [M]$, denote $\mathbf{a}_{i0}' = \mathbf{a}_i'$
10:     **end for**
11:     Calculate $\mathcal{L}_Q(\boldsymbol{\psi}) = \sum_i \left\| Q^\psi(\mathbf{x}_i, \mathbf{a}_i) - r_i - \gamma \frac{\sum_{j=0}^M \exp(\beta Q_\perp^\psi(\mathbf{x}_i', \mathbf{a}_{ij}')) Q_\perp^\psi(\mathbf{x}_i', \mathbf{a}_{ij}')}{\sum_{j=0}^M \exp(\beta Q_\perp^\psi(\mathbf{x}_i', \mathbf{a}_{ij}'))} \right\|_2^2$
12:     Update $\boldsymbol{\psi}$ using the gradient of $\mathcal{L}_Q(\boldsymbol{\psi})$
13: **end for**
14: **for** policy improvement step $k \in [K_3]$ **do**
15:     **for** batch $\{(\mathbf{x}_i, \mathbf{a}_i, \mathbf{x}_i', r_i)\}_i \subset \mathcal{D}$ **do**
16:         **if** $k \bmod K_{\text{renew}} = 1$ **then**
17:             Sample support action set $\mathbf{a}_{ij}$ using vector field $\mathbf{v}^{\boldsymbol{\theta}}(\cdot|\mathbf{x}_i)$ for all $i \in [B], j \in [M]$
18:             Denote $\mathbf{a}_{i0} = \mathbf{a}_i$, sample $t_{ij} \sim U(0,1)$ and $\mathbf{a}_{ij,t} \sim p_{t_{ij}0}(\cdot|\mathbf{a}_{ij})$ for all $j \in \overline{[M]}$
19:             Calculate guidance $g_{ij} = \text{softmax}_j(\beta Q^\psi(\mathbf{x}_i, \mathbf{a}_{ij})) = \frac{\exp(\beta Q^\psi(\mathbf{x}_i, \mathbf{a}_{ij}))}{\sum_{j=0}^M \exp(\beta Q^\psi(\mathbf{x}_i, \mathbf{a}_{ij}))}$
20:         **end if**
21:         Calculate loss $\mathcal{L}_{\text{QF}}(\boldsymbol{\theta}) = \sum_{i=1,j=0}^{B,M} \lambda(t_{ij}) g_{ij} \left\| \mathbf{v}_{t_{ij}}^{\boldsymbol{\theta}}(\mathbf{a}_{ij,t_{ij}}; \mathbf{x}_i) - \mathbf{u}_{t_{ij}0}(\mathbf{a}_{ij,t_{ij}}|\mathbf{a}_{ij}) \right\|_2^2$
22:         Update $\boldsymbol{\theta}$ using the gradient of $\mathcal{L}_{\text{QF}}(\boldsymbol{\theta})$
23:     **end for**
24: **end for**

---

---

**Algorithm 5** Q-weighted iterative policy optimization for offline RL (Diffusion)

---

**Input:** Score function $\mathbf{s}_t^{\boldsymbol{\theta}}(\cdot)$, guidance scale $\beta$, batch size $B$, weight on time $\lambda(t)$, schedule $(\mu_t, \sigma_t)$
**Input:** Offline RL dataset $\mathcal{D} = \{(\mathbf{x}, \mathbf{a}, \mathbf{x}', r)\}$, number of epochs $K_1, K_2$ and $K_3$, Q function $Q^{\psi}$
**Input:** Support action set size $M$, support action set renew frequency $K_{\mathrm{renew}}$

1: **for** diffusion model warm-up epoch $k \in [K_1]$ **do**
2:     **for** batch $\{(\mathbf{x}_i, \mathbf{a}_i, \mathbf{x}'_i, r_i)\}_i \subset \mathcal{D}$ **do**
3:         Sample $t_i \sim U(0, 1)$, calculate $\mu_{t_i}, \sigma_{t_i}$, sample $\boldsymbol{\epsilon}_i \sim \mathcal{N}(0, \mathbf{I}_d)$ and $\mathbf{a}_{i,t_i} = \mu_{t_i}\mathbf{a}_i + \sigma_{t_i}\boldsymbol{\epsilon}_i$
4:         Calculate loss $\mathcal{L}_{\mathrm{QD, \, warmup}}(\boldsymbol{\theta}) = \sum_{i=1}^B \lambda(t_i) \|\mathbf{s}_t^{\boldsymbol{\theta}}(\mathbf{a}_{i,t}; \mathbf{x}_i) + \boldsymbol{\epsilon}_i\|_2^2$
5:         Update $\boldsymbol{\theta}$ using the gradient of $\mathcal{L}_{\mathrm{QD, \, warmup}}(\boldsymbol{\theta})$
6:     **end for**
7: **end for**
8: **for** Q-learning warm-up epoch $k \in [K_2]$ **do**
9:     **for** batch $\{(\mathbf{x}_i, \mathbf{a}_i, \mathbf{x}'_i, r_i)\}_i \subset \mathcal{D}$ **do**
10:        Sample support action set $\mathbf{a}'_{ij}$ using score function $\mathbf{s}_t^{\boldsymbol{\theta}}$ for $j \in [M]$, denote $\mathbf{a}'_{i0} = \mathbf{a}'_i$
11:     **end for**
12:     Calculate $\mathcal{L}_Q(\boldsymbol{\psi}) = \sum_i \left\| Q^{\psi}(\mathbf{x}_i, \mathbf{a}_i) - r_i - \gamma \frac{\sum_{j=0}^M \exp(\beta Q_{\perp}^{\psi}(\mathbf{x}'_i, \mathbf{a}'_{ij})) Q_{\perp}^{\psi}(\mathbf{x}'_i, \mathbf{a}'_{ij})}{\sum_{j=0}^M \exp(\beta Q_{\perp}^{\psi}(\mathbf{x}'_i, \mathbf{a}'_{ij}))} \right\|_2^2$
13:     Update $\boldsymbol{\psi}$ using the gradient of $\mathcal{L}_Q(\boldsymbol{\psi})$
14: **end for**
15: **for** policy improvement step $k \in [K_3]$ **do**
16:     **for** batch $\{(\mathbf{x}_i, \mathbf{a}_i, \mathbf{x}'_i, r_i)\}_i \subset \mathcal{D}$ **do**
17:        **if** $k \bmod K_{\mathrm{renew}} = 1$ **then**
18:           Sample support action set $\mathbf{a}_{ij}$ using score function $\mathbf{s}^{\boldsymbol{\theta}}(\cdot | \mathbf{x}_i)$ for all $i \in [B], j \in [M]$
19:           Denote $\mathbf{a}_{i0} = \mathbf{a}_i$, sample $t_{ij} \sim U(0, 1)$, calculate $\mu_{t_{ij}}, \sigma_{t_{ij}}$
20:           Sample $\boldsymbol{\epsilon}_{ij} \sim \mathcal{N}(0, \mathbf{I}_d)$ for $i \in [B], j \in [M]$, let $\mathbf{a}_{ij,t_{ij}} = \mu_{t_{ij}}\mathbf{a}_{ij} + \sigma_{t_{ij}}\boldsymbol{\epsilon}_{ij}$
21:           Calculate guidance $g_{ij} = \mathrm{softmax}_j(\beta Q^{\psi}(\mathbf{x}_i, \mathbf{a}_{ij})) = \frac{\exp(\beta Q^{\psi}(\mathbf{x}_i, \mathbf{a}_{ij}))}{\sum_{j=0}^M \exp(\beta Q^{\psi}(\mathbf{x}_i, \mathbf{a}_{ij}))}$
22:        **end if**
23:        Calculate loss $\mathcal{L}_{\mathrm{QD}}(\boldsymbol{\theta}) = \sum_{i=1, j=0}^{B, M} \lambda(t_{ij}) g_{ij} \|\mathbf{s}_{t_{ij}}^{\boldsymbol{\theta}}(\mathbf{a}_{ij,t_{ij}}; \mathbf{x}_i) + \boldsymbol{\epsilon}_{ij}/\sigma_{t_{ij}}\|_2^2$
24:        Update $\boldsymbol{\theta}$ using the gradient of $\mathcal{L}_{\mathrm{QD}}(\boldsymbol{\theta})$
25:     **end for**
26: **end for**

---

# E   ADDITIONAL EXPERIMENT RESULTS

We first present the comparison between QIPO and other algorithm in Table 4.

## E.1   EXPERIMENT CONFIGURATIONS

**QIPO-Diff configurations** We adopt the same network structure and training configuration as Lu et al. (2023) during the score network warmup $K_1$ and Q-function training $K_2$ for a fair comparison and to ensure consistency across experiments. We finetune the score network $\mathbf{s}_t^{\boldsymbol{\theta}}$ obtained after warm-up (Line 5, Algorithm 5) with learning rate $10^{-4}$ to perform the Q-weighted iterative policy optimization. The schedule of the diffusion is the same with Lu et al. (2023) and we use the DPM-Solver (Lu et al., 2022a) with `diffusion-step = 15` to accelerate the generation step, which is also the same as (Lu et al., 2023).

**QIPO-OT configurations** We use the optimal transport VF (Lipman et al., 2022) defined by

$$\mathbf{u}_{t0}(\mathbf{x}|\mathbf{x}_0) = \frac{\mathbf{x}_0 - (1 - \sigma_{\min})\mathbf{x}}{1 - (1 - \sigma_{\min})t}, \sigma_{\min} = 0.0054,$$

note we switched the notation between $t = 0$ and $t = 1$ compared with (Lipman et al., 2022). According to Lemma 3.2, we defined the network of $\mathbf{v}_t^{\boldsymbol{\theta}}$ as

$$\mathbf{v}_t^{\boldsymbol{\theta}}(\mathbf{x}) = (1 - t)^{-1}\log t\mathbf{x} + (1 - t)^{-1}t\left(t\log t - (1 - t)^{-1}\right)\mathbf{v}_t^{\boldsymbol{\theta}}(\mathbf{x}),$$

where $\mathbf{s}_t^{\boldsymbol{\theta}}$ is the same with QIPO-Diff implementation. Similar with QIPO-Diff, we employ the DPM-Solver (Lu et al., 2022a) for the second-order Runge–Kutta–Fehlberg method for solving the reverse-time ODE $\frac{\mathrm{d}\mathbf{x}}{\mathrm{d}t} = \mathbf{v}_t^{\boldsymbol{\theta}}(\mathbf{x})$.

Table 4: Evaluation numbers of D4RL benchmarks (normalized as suggested by Fu et al. (2020)). We report mean $\pm$ standard deviation of algorithm performance across 8 random seeds. The result for ablation study with $\beta = 10$ are the averaged performance over 3 random seeds. Together with the benchmark algorithms presented in Table 2, the highest performance is **boldfaced highlighted.** The performance within 5% of the maximum absolute value in every individual task are highlighted. To ensure fairness, ablation results with $\beta = 10$ are excluded from the best performance comparison. However they are still highlighted if they meet the 5% performance criterion.

| Dataset | Environment | BEAR | TD3+BC | IQL | Diffuser | Diffusion-QL | QIPO-Diff (ours) | QIPO-Diff (ours, $\beta = 10$) |
|---|---|---|---|---|---|---|---|---|
| Medium-Expert | HalfCheetah | 53.4 | 90.7 | 86.7 | 79.8 | 96.8 | $94.14 \pm 0.48$ | 94.96 |
| Medium-Expert | Hopper | 96.3 | 98.0 | 91.5 | 107.2 | 111.1 | $\mathbf{112.12 \pm 0.42}$ | 103.56 |
| Medium-Expert | Walker2d | 40.1 | 110.1 | 109.6 | 108.4 | 110.1 | $110.14 \pm 0.51$ | 112.09 |
| Medium | HalfCheetah | 41.7 | 48.3 | 47.4 | 44.2 | 51.1 | $48.19 \pm 0.20$ | 53.11 |
| Medium | Hopper | 52.1 | 59.3 | 66.3 | 58.5 | 90.5 | $89.53 \pm 9.96$ | 92.01 |
| Medium | Walker2d | 59.1 | 83.7 | 78.3 | 79.7 | 87.0 | $84.99 \pm 0.46$ | 86.36 |
| Medium-Replay | HalfCheetah | 38.6 | 44.6 | 44.2 | 42.2 | 47.8 | $45.27 \pm 0.42$ | 50.10 |
| Medium-Replay | Hopper | 33.7 | 60.9 | 94.7 | 101.3 | 100.7 | $101.23 \pm 0.47$ | 99.57 |
| Medium-Replay | Walker2d | 19.2 | 81.8 | 73.9 | 61.2 | 95.5 | $90.08 \pm 4.53$ | 89.45 |
| **Average (Locomotion)** | | 51.9 | 75.3 | 76.9 | 75.3 | 88.0 | 86.20 | 86.80 |
| Default | AntMaze-umaze | 73.0 | 78.6 | 87.5 | - | 93.4 | $\mathbf{97.5 \pm 0.53}$ | 98.33 |
| Diverse | AntMaze-umaze | 61.0 | 71.4 | 62.2 | - | 66.2 | $73.88 \pm 6.42$ | 83.00 |
| Play | AntMaze-medium | 0.0 | 10.6 | 71.2 | - | 76.6 | $82.75 \pm 3.24$ | 92.67 |
| Diverse | AntMaze-medium | 8.0 | 3.0 | 70.0 | - | 78.6 | $\mathbf{86.00 \pm 8.65}$ | 89.00 |
| Play | AntMaze-large | 0.0 | 0.2 | 39.6 | - | 46.4 | $\mathbf{73.25 \pm 10.90}$ | 62.67 |
| Diverse | AntMaze-large | 0.0 | 0.0 | 47.5 | - | 56.6 | $40.5 \pm 20.40$ | 59.00 |
| **Average (AntMaze)** | | 23.7 | 27.3 | 63.0 | - | 69.6 | 77.3 | 80.78 |

## E.2 INDIVIDUAL TIMING COMPARISON BETWEEN QGPO AND QIPO

We present the timing comparison between QGPO (Lu et al., 2023) and QIPO in this section, as shown in Figure 4. It is evident that QIPO-OT consistently outperforms QIPO-Diff in terms of speed, with QGPO being the slowest due to the additional back-propagation. The advantage of optimal-transport flow matching over diffusion models aligns with the claims made by Lipman et al. (2022) and Zheng et al. (2023), which suggest that OT can accelerate the generation process.

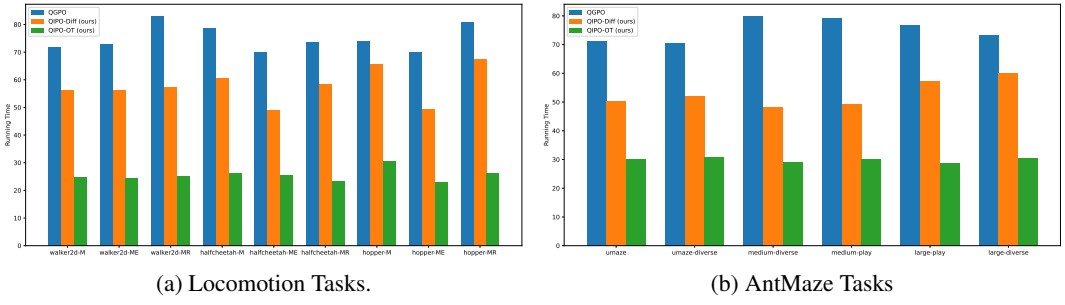

(a) Locomotion Tasks.             (b) AntMaze Tasks

Figure 4: Comparison of the time complexity of the action selecting function for CEP, QIPO-Diff and QIPO-OT across various tasks. The results are averaged over 3 individual runs.

**Iterative Policy Optimization** We perform the iterative policy optimization (Line 7, Algorithm 2) with $K_3 = 100$ and evaluate the performance of the agent every 5 epochs. We renew the support set with period $K_{\text{renew}} = 10$. The rewards reported in Table 2 and Table 4 are the best performance of the reported performance over $K_3 = 100$ episodes. Since the QIPO procedure progressively improve the guidance scale beta, as discussed in (5.4), this is similar to automatically tune the guidance scale $s$ in Lu et al. (2023) as a hyper-parameter. We use the soft update (Lillicrap, 2015) with $\lambda = 0.005$ to stabilize the update of the score network,

## E.3 ABLATION STUDIES FOR QIPO

**Changing the guidance scale** $\beta$. We conduct an ablation study by changing the guidance scale from $\beta = 3$ to $\beta = 10$ for QIPO-Diff, with the results included in the last column of Table 4.

The algorithm demonstrates robustness to changes in the guidance scale, as the iterative policy optimization process naturally adapts and improves the guidance. Additionally, we observed that the average performance over 3 random seeds sometimes surpasses the original setting with $\beta = 3$, while in other cases it is slightly below the original performance.

**Changing the support action set $M$.** We ablate the size of the support action set, increasing it from $M = 16$ to $M = 32$ and $M = 64$, as shown in Figure 5. The results suggest that a larger support set $M$ may lead to slightly improved performance. However, since sampling the support action set requires running the reverse process of the diffusion model or the score matching model, we choose to keep $M = 16$ to balance efficiency and effectiveness.

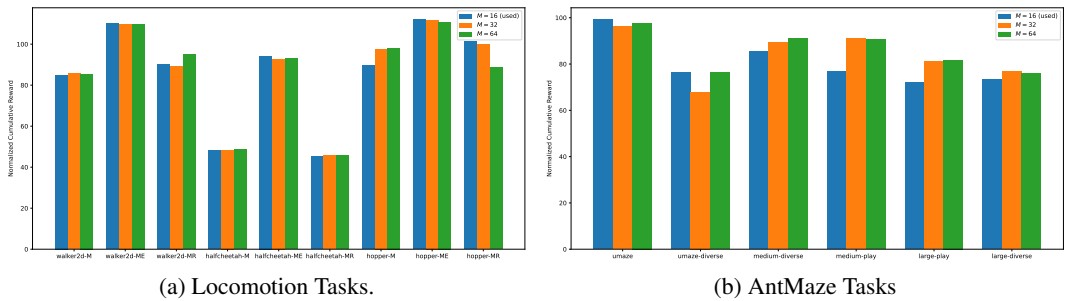

(a) Locomotion Tasks.          (b) AntMaze Tasks

Figure 5: Comparison of normalized cumulative rewards for different support set sizes ($M = 16$, $M = 32$, and $M = 64$) across various tasks. The results are averaged over 3 individual runs.

**Changing the renew support action period $K_{\text{renew}}$.** We ablate the period for changing the support action set within $K_{\text{renew}} \in \{5, 10, 20, 50\}$. Intuitively, a larger $K_{\text{renew}}$ may result in a better-learned score network but slower updates to the guidance scale. As shown in Figure 6, we observe that smaller $K$ values generally lead to improved performance. We hypothesize that this is because the score network for the D4RL tasks is relatively easy for the neural network to learn.

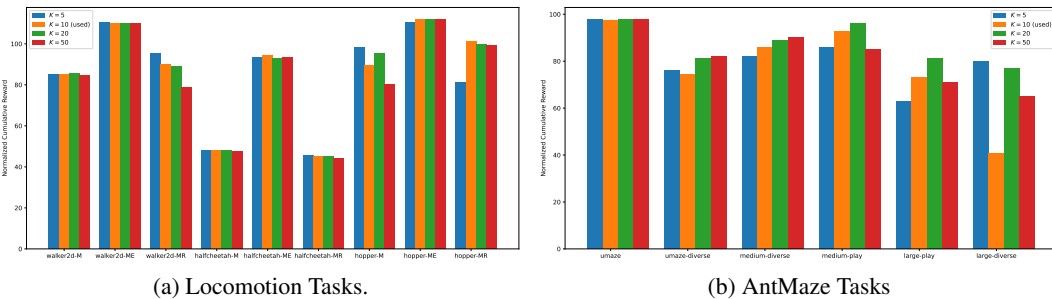

(a) Locomotion Tasks.          (b) AntMaze Tasks

Figure 6: Comparison of normalized cumulative rewards for different support action set renewal period ($K_{\text{renew}} \in \{5, 10, 20, 50\}$) across various tasks. The results are averaged over 3 individual runs.

## F  ADDITIONAL EXPERIMENTS

### F.1  CHANGING THE ENERGY LANDSCAPE

We conduct an ablation study to analyze the performance of the energy-weighted diffusion model under varying energy landscapes. Specifically, we use the 8-Gaussian example shown in Figure 3b and modify the energy landscape as illustrated in Figure 7a. The sampling results obtained from running Algorithm 1 are depicted in Figure 7b. These results suggest that the proposed energy-weighted diffusion model is robust to changes in the energy landscape.

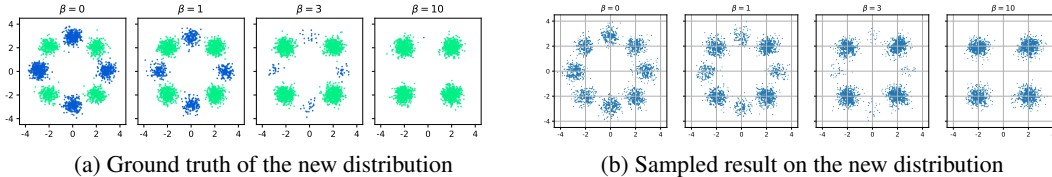

(a) Ground truth of the new distribution      (b) Sampled result on the new distribution

Figure 7: Ground truth and sampling results on the 8-Gaussian dataset with a modified energy landscape.

### F.2   TAKE $\beta$ AS A SCORE NETWORK INPUT

Although Algorithms 3 and 1 require the guidance scale $\beta$ as an input for simplicity of presentation, we propose an additional algorithm that unifies different choices of $\beta$ across the range from 0 to $\beta_{\max}$. As described in Algorithm 6, this algorithm treats $\beta$ as a network input and, for each minibatch, randomly selects a $\beta$ value from the uniform distribution $U(0, \beta_{\max})$. The encoding of $\beta$ follows the same method used for encoding time $t$ in the network, represented as:

$$\widetilde{\mathbf{x}} = \mathbf{x} \oplus \mathrm{emb}(t) \oplus \mathrm{emb}(\beta/\beta_{\max}),$$

where $\mathrm{emb}(\cdot)$ denotes the time embedding function and $\oplus$ indicates concatenation. The reverse (generation) process remains unchanged, except that the score function is modified to $\mathbf{s}^{\boldsymbol{\theta}}(\cdot; \cdot, \beta)$, using a specific guidance scale $\beta$. We conducted experiments on the bandit case illustrated in Figure 8. The results indicate that the performance of the proposed algorithm is comparable to that of Algorithms 3 and 1, suggesting that the score model can effectively learn multiple $\beta$ values simultaneously.

---

**Algorithm 6** Training Energy-Weighted Diffusion Model with $\beta$ as Input

---

**Input:** Score function $\mathbf{s}\boldsymbol{\theta}(\cdot; \cdot, \cdot)$, maximum guidance scale $\beta_{\max}$
**Input:** Batch size $B$, time weight $\lambda(t)$, schedule $(\mu_t, \sigma_t)$
1: **for** batch $\{\mathbf{x}_0^i, \mathcal{E}(\mathbf{x}_0^i)\}_i$ **do**
2:      Sample $\beta \sim U(0, \beta_{\max})$
3:      **for** index $i \in [B]$ **do**
4:          Calculate guidance $g_i = \mathrm{softmax}(-\beta \mathcal{E}(\mathbf{x}_0^i)) = \exp(-\beta \mathcal{E}(\mathbf{x}_0^i)) / \sum_j \exp(-\beta \mathcal{E}(\mathbf{x}_0^j))$
5:          Sample $t_i \sim U(0,1)$, calculate $\mu_{t_i}, \sigma_{t_i}$, sample $\boldsymbol{\epsilon}_i \sim \mathcal{N}(0, \mathbf{I}_d)$ and $\mathbf{x}_{t_i} = \mu_{t_i} \mathbf{x}_0^i + \sigma_{t_i} \boldsymbol{\epsilon}_i$
6:      **end for**
7:      Calculate and take a gradient step using $\mathcal{L}_{\mathrm{CED}}(\boldsymbol{\theta}) = \sum_i \lambda(t_i) g_i \|\mathbf{s}^{\boldsymbol{\theta}}(\mathbf{x}_{t_i}; t_i, \beta) + \boldsymbol{\epsilon}_i / \sigma_{t_i}\|_2^2$.
8: **end for**

---

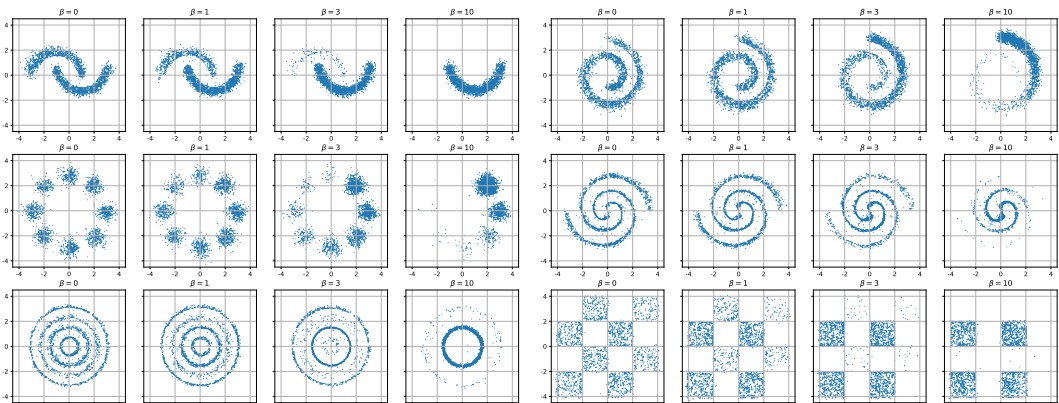

Figure 8: Data sampled from the score function trained using energy-weighted diffusion with $\beta$ as an input. A total of 2,000 data points were sampled for each state.

### F.3  Image Synthesis

To evaluate the performance of the proposed methods in guiding the generative process for more complex tasks, we conduct image synthesis experiments using the `smithsonian-butterflies-subset` dataset[3]. Similar to Lu et al. (2023), the energy function is defined as the average hue distance between the image and the target hues red, blue, and green, expressed as $\mathcal{E}(\mathbf{x}) = \|h(\mathbf{x}) - h_{\text{target}}\|_1 \in [0, 1]$. We use the energy-weighted diffusion to train for the target distribution described as $q(\mathbf{x}) \propto p(\mathbf{x}) \exp(-\beta \mathcal{E}(\mathbf{x}))$ with $\beta = 50$ in the experiments. The results, presented in Figure 9, demonstrate that our methods effectively guide the image generation process.

### F.4  Molecular Structure Generation

We further evaluate the performance of the proposed weighted diffusion algorithms on molecular structure generation using E(3) diffusion networks (Hoogeboom et al., 2022). The normalized properties—namely, isotropic polarizability ($\alpha$), highest occupied molecular orbital energy ($\epsilon_{\text{HOMO}}$), lowest unoccupied molecular orbital energy ($\epsilon_{\text{LUMO}}$), and the energy gap $\Delta_\epsilon$ between $\epsilon_{\text{HOMO}}$ and $\epsilon_{\text{LUMO}}$—are used to guide the generation process. For instance, when the energy function is set as $E(\mathbf{x}) = \alpha$, the diffusion model is guided to generate molecular structures with higher isotropic polarizability. The energy-guided is target for generating the distribution $q(\mathbf{x}) \propto p(\mathbf{x}) \exp(\beta \mathcal{E}(\mathbf{x}))$ in this case. Similar to (Hoogeboom et al., 2022), we train a separate EGNN (Satorras et al., 2021) to predict the normalized properties of the generated molecules. The property normalization method follows (Hoogeboom et al., 2022), and we report the performance for $\beta \in 0, 1, 3, 10$ in Table 5. The results indicate that the energy-weighted diffusion model can successfully generate molecular structures with enhanced desired properties.

| Property | $\beta = 0$ | $\beta = 1$ | $\beta = 3$ | $\beta = 10$ | **error** |
|:---:|:---:|:---:|:---:|:---:|:---:|
| $\alpha$ | $-0.1977$ | $0.6196$ | $2.2965$ | $2.2956$ | $0.1034$ |
| $\Delta_\epsilon$ | $-0.1910$ | $0.4687$ | $1.7025$ | $1.7129$ | $0.0667$ |
| $\epsilon_{\text{HOMO}}$ | $0.1142$ | $1.6074$ | $2.7396$ | $2.6798$ | $0.0413$ |
| $\epsilon_{\text{LUMO}}$ | $-0.2150$ | $0.4089$ | $1.7605$ | $1.7669$ | $0.0362$ |

Table 5: Averaged predicted normalized properties for molecular generation. The values are averaged over 1,000 generated molecular structures. The column **error** represents the prediction error of the classifier. All properties are normalized by their mean and standard deviation from the training dataset.

### F.5  Training Curves of OT and Diffusion

In this subsection we plot the training curves for learning the behavioral policy $\mu(a|s)$ following the flow matching (OT) and diffusion loss to further demonstrate the comparison between OT and diffusion when applied in offline RL tasks. In particular, Figure 10 compares the diffusion and OT training loss on the 6 ant-maze tasks. Since it is impossible to find a $\mathbf{v}_{\boldsymbol{\theta}}^t(\mathbf{a}_t; \mathbf{x})$ (or $\mathbf{s}_{\boldsymbol{\theta}}^t(\mathbf{a}_t; \mathbf{x})$ matching all $\mathbf{u}_{t0}(\mathbf{a}_t|\mathbf{a}_0)$ (or $\nabla_{\mathbf{a}_t} \log p_{t0}(\mathbf{a}_t|\mathbf{a}_0)$) for all $\mathbf{a}_0$, neither the flow matching loss $\|\mathbf{v}_{\boldsymbol{\theta}}^t(\mathbf{a}_t; \mathbf{x}) - \mathbf{u}_{t0}(\mathbf{a}_t|\mathbf{a}_0)\|_2^2$ nor the score matching (diffusion) loss $\|\mathbf{s}_{\boldsymbol{\theta}}^t(\mathbf{a}_t; \mathbf{x}) - \nabla_{\mathbf{a}_t} \log p_{t0}(\mathbf{a}_t|\mathbf{a}_0)\|_2^2$ converge to zero, and the remaining training loss is different between OT and diffusion. Therefore we normalize the loss and presented in Figure 11 by shifting the loss at the last epoch (600-th epoch) as 0 and the loss at the first epoch as 1.

The experiment results as presented in Figure10, 11 suggest the following findings. First, on the Antmaze tasks, flow matching using OT loss converges similar but slightly faster to the stable point. Second, despite of this difference on the convergence rate, both OT and diffusion can correctly learn the behavioral policy.

---

[3]https://huggingface.co/datasets/huggan/smithsonian_butterflies_subset

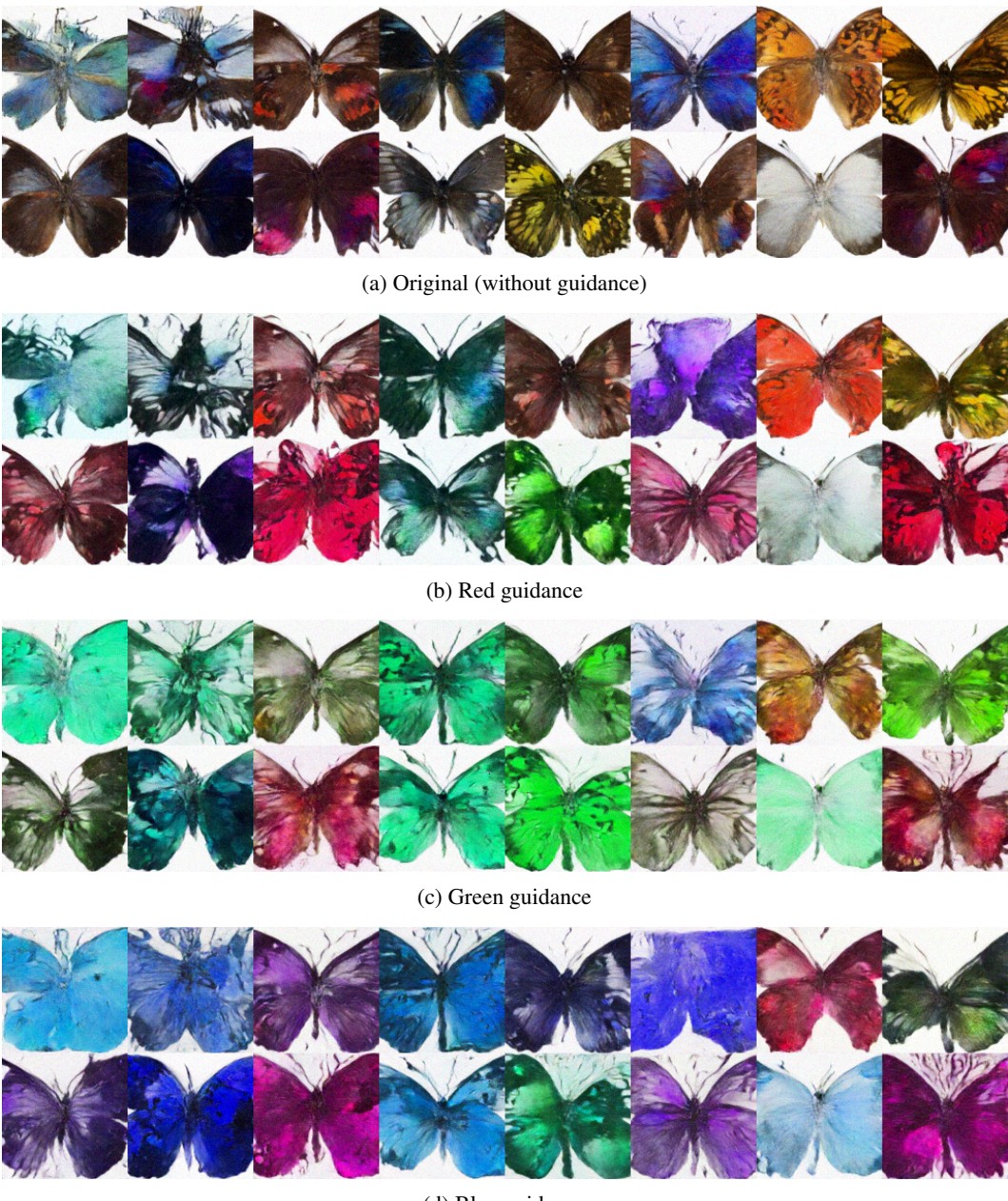

(a) Original (without guidance)

(b) Red guidance

(c) Green guidance

(d) Blue guidance

Figure 9: Generated butterflies, the samples are generated from the same 16 random seeds with different guidance or no guidance using reverse process with $T = 1000$

.

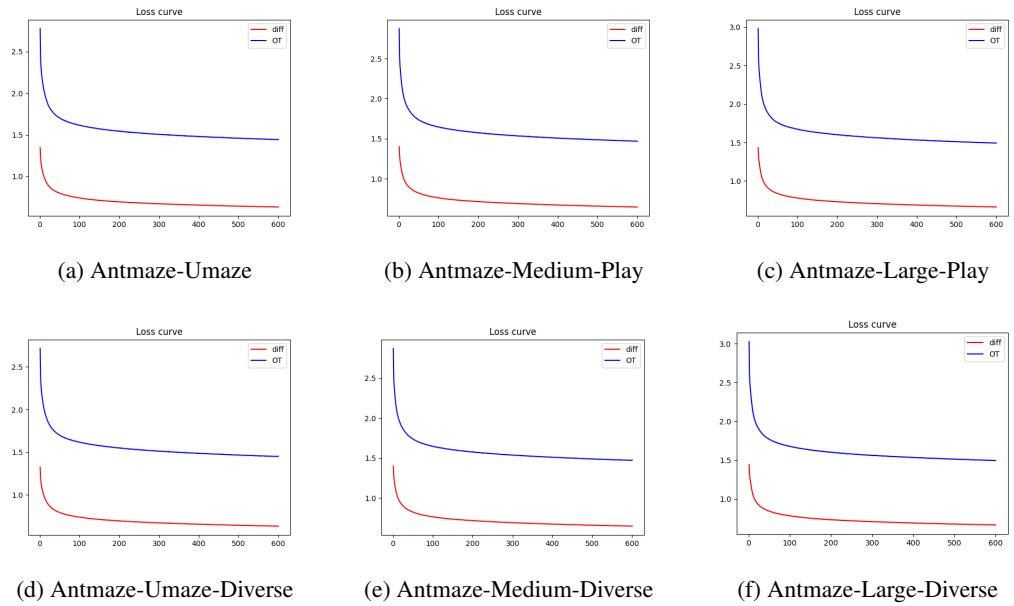

Figure 10: Unnormalized loss w.r.t. number of epochs for the first 600 epochs in learning the behavioral policy $\mu$. The blue curve indicates the OT schedule for flow matching and the red curve indicates the diffusion schedule.

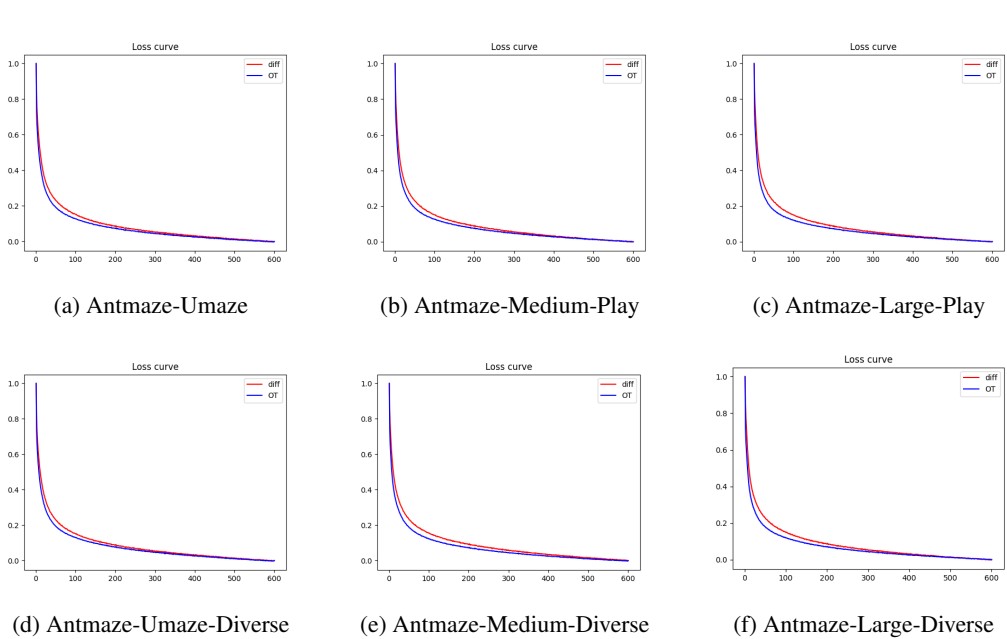

Figure 11: Normalized loss w.r.t. number of epochs for the first 600 epochs in learning the behavioral policy $\mu$. The blue curve indicates the OT schedule for flow matching and the red curve indicates the diffusion schedule. The normalization is conducted by rescaling the loss at last iteration to 0 and the loss at the first iteration to 1.

