# OpenReview forum: "Energy-Weighted Flow Matching for Offline Reinforcement Learning"
_ICLR.cc/2025/Conference — ICLR 2025 Poster_

### Official Review · Reviewer_JYA4 · 2024-10-26

**Soundness:** 3
**Presentation:** 2
**Contribution:** 2
**Rating:** 6
**Confidence:** 4

**Summary:**

This manuscript discussed how to estimate the score function with an energy guidance without axillary model but can still maintain the exact guidance. The main idea is decomposing the supervision from the data population and energy guidance. The authors also extended the proposed methods to KL-regularized offline reinforcement learning.

**Strengths:**

* The derivations are sound and the final algorithm is intuitive.

**Weaknesses:**

* The presentation is not so straightforward.
* If I understand the proposal correctly, for each $\beta$ the proposed method needs to at least fine tune the score function and do the sampling. I don’t think it will have significant benefits on we directly estimate the score function without leveraging the proposed methods.

**Questions:**

* Will Monte-Carlo estimation on the denominator lead to a high variance on the loss/gradient estimator? It would be generally not a good idea to do something like this.
* Do CG/CFG and CEP/proposed methods only differ from the diffusion path and do we have an understanding on why this two diffusion path will lead to different empirical results?

---

### Official Review · Reviewer_mBzj · 2024-11-01

**Soundness:** 2
**Presentation:** 2
**Contribution:** 2
**Rating:** 6
**Confidence:** 3

**Summary:**

This paper presents a novel diffusion method combined with offline RL. The major contribution is that it combines flow matching loss with an energy function, specifically weighting the flow matching term with the regularized energy, which avoids calculating the gradient of the intermediate energy function. This idea is integrated into offline RL, leading to the development of Q-weighted iterative policy optimization (QIPO) with two variant velocity fields, OT and VP-SDE. Experiments on toy examples reveal the effectiveness of ED and QIPO on D4RL datasets compared to SOTA methods.

**Strengths:**

1. The authors provide substantial theoretical analysis.

2. The experiments reveal that QIPO-Diff and QIPO-OT outperform current SOTA offline RL methods.

**Weaknesses:**

1.The authors should discuss more details on the benefits of QIPO. The current methods only discuss the advantage of ED and CED over CEP, which is mainly about the diffusion models. It seems that the advantage of QIPO over other offline RL methods (e.g., SRPO, SfBC) is missing.


2. Only toy experiments on ED are given. If the authors want to claim the advantages of ED and CED, more experiments should be included, e.g., image synthesis tasks on ImageNet. I understand that verification of the diffusion model in different tasks might not be a major topic in this paper, but in its current form, QIPO appears to be a direct application of ED in the offline RL setting due to ED's advantages. Thus, empirical verification of ED is important.



3. The theoretical analysis should add more details and fix typos, e.g., in Eq. (a.5), how the div operator is eliminated should be discussed in detail.

**Questions:**

Please answer my questions mentioned above.

---

> ### Author Response · Authors · 2024-11-20
> **Response to Reviewer mBzj**
>
> Thank you for your feedback. Here we address your questions.
>
> **Q1. What’s the advantage of the proposed method compared with SRPO and SfBC?**
>
> **A1.** The advantage of our proposed method over SRPO and SfBC lies in its formulation as an energy-guided generative model that samples from the distribution $q(x) \propto p(x) \exp(-\beta E(x))$. In contrast:
> - **SRPO** uses the diffusion model primarily as a regularization mechanism for policy optimization, focusing on maximizing the $Q$ function (which acts as the energy function in this context).
> - **SfBC** employs the diffusion model as a data augmentation tool during the planning phase.
>
> Both SRPO and SfBC are limited in that they cannot be extended to more general settings where both maximizing the $Q$ function and aligning with the data distribution are equally important. Our method, however, effectively balances these two objectives, making it suitable for a wider range of generative tasks.
>
> **Q2. Can this method be extended to more general tasks?**
>
> **A2.** We appreciate your insights on the potential extensions of our algorithm beyond RL. In response, we conducted experiments on image synthesis and molecular generation, following established benchmarks (DDPM [1], EDM[2]). We have included results in **Appendix F.3** and **F.4**. These results indicate that energy-weighted diffusion can be readily applied to guide complex generative tasks, where standard RL algorithms may not be effective.
>
> **Q3. How the div operator is eliminated**
>
> **A3.** Apologies for the typo. We intended to write $\dot{q_t}(x) = -\mathrm{div} \cdot [q_t(x) \hat{u}_t(x)]$. This has been corrected in the revised manuscript.
>
> [1] Ho, Jonathan, Ajay Jain, and Pieter Abbeel. "Denoising diffusion probabilistic models." Advances in neural information processing systems 33 (2020): 6840-6851.
>
> [2] Hoogeboom, Emiel, et al. "Equivariant diffusion for molecule generation in 3d." International conference on machine learning. PMLR, 2022.

---

> > ### Author Response · Authors · 2024-11-22
> > **Follow up with Reviewer mBzj**
> >
> > Thank you for your constructive questions and feedback! We’d like to follow up to ensure that our response has addressed your concerns. In our previous response, we:
> > - Discussed the connection and differences between our method (energy-guided generation) and SfBC/SRPO. A key distinction is that SfBC and SRPO are not capable of guiding the generation process, making them unsuitable for tasks like image synthesis.
> > - Provided experiments on image synthesis tasks (Appendix F.3) and molecular generation tasks (Appendix F.4).
> > - Correct a typo you pointed out
> >
> > Additionally, we’ve uploaded a revised version with more experiments to further clarify our approach. We hope this addresses your concerns and look forward to continuing the discussion to answer any remaining questions you might have.

---

> > > ### Comment · Reviewer_mBzj · 2024-11-27
> > >
> > > Thank you for the authors' feedback. Many of my concerns have been addressed properly, and I have accordingly increased my score.
> > >
> > > However, I have a follow-up question regarding Q1. I understand the explanation of the advantage of the proposed method, both maximizing the Q function (energy function) and aligning with the data distribution, in RL. However, since the proposed method is a relatively general energy-guided diffusion framework, could the explanation be extended to other scenarios, such as image synthesis? Specifically, in the proposed image synthesis setting (sec. F.3), the energy function is based on the average hue distance (line 1356). In this case, maximizing the energy function does not seem appropriate, as the goal is to minimize the distance. Could the authors elaborate further on this point?

---

> ### Author Response · Authors · 2024-11-27
> **Thank you for your feedback!**
>
> Thank you for your feedback and for updating your recommendation!
>
> Regarding your question on whether the energy is being maximized or minimized, we would like to clarify that the difference arises from the specific definitions of the target distribution in different settings.
>
> In the general setting discussed in Section 4, we define the target distribution as  $q(x) \propto p(x) \exp(-\beta E(x))$ . Here, when the energy function is defined as the hue value distance, the energy guidance will direct the generative model to find images with smaller hue distances to the target color.
>
> In contrast, in RL, the target policy is defined as $\pi(a | s) \propto \mu(a | s) \exp(\beta Q(s, a))$ . In this case, the reviewer is correct that the guidance seeks to maximize the  $Q$-function. However, this difference is merely a matter of the sign of the energy term:  $-\beta E(x)$  corresponds to minimizing energy, while  $\beta E(x)$  corresponds to maximizing energy. The only thing changed is the sign before the energy function used in CWD training process.
>
> To address this explicitly, we have clarified in the newly uploaded revision that for each application—molecular generation and image synthesis—the energy function is defined consistently with the target distribution. Specifically:
> - Image generation: The energy function is defined as the hue distance, which is minimized via  $q(x) \propto p(x) \exp(-\beta E(x))$ to guide the generation to a certain color.
> - Molecular conformation synthesis: The energy function corresponds to desired molecular properties, which are maximized via  $q(x) \propto p(x) \exp(\beta E(x))$.
>
> We hope this explanation addresses your question.

---

### Official Review · Reviewer_6Qqj · 2024-11-02

**Soundness:** 4
**Presentation:** 3
**Contribution:** 4
**Rating:** 8
**Confidence:** 3

**Summary:**

This paper proposes a training scheme for energy-guided continuous generative models, i.e. models that learn $q(x)\propto p(x)\exp(-E(x))$. To do so it leverages something akin to the $x_0$-conditional trick of Lipman to rewrite a loss that would depend on an intermediate energy $E(x_t)$ as simply depending on $E(x_0)$. Functionally, this is essentially a softmax-of-energy weighted loss for both flow matching and score matching models. This provably converges in the limit to an unbiased model that, unlike prior work, does not require a second model to learn a guidance.

The authors demonstrate their method on simple demonstrative tasks and a set of benchmark offline-RL continuous control tasks, where the method performs on-par with state of the art methods.

**Strengths:**

The paper proposes a novel solution to a very important problem, and the solution itself is fairly elegant (as most unbiased methods are in this reviewer's opinion). The paper was easy to read and used an appropriate level of detail to explain and illustrate the method. The empirical work also shows good performance on non-trivial problems.

**Weaknesses:**

The main weakness of the paper is it doesn't really go into the weeds of the method. Unbiased methods are great, but sometimes they come at a cost. For example, importance sampling methods are commonly thought of as high variance. It's not clear if this is the case here, or more generally what are the trade offs that would inform choosing this method over others.

I know it's easy to write this, but there could be more empirical work done. Specifically, two very common class of problems for which these methods are used are missing, image generation and physics problems (e.g. 3d molecular conformer generation). RL is a challenging problem but the method is much more general than that, and so it feels like a missed opportunity that results on these other problems are not presented.

**Questions:**

One of the assumptions made seems to be that there is a prior (data distribution) $p_0(x_0)$ we can sample from, but results will differ greatly if $p_0$ is an actual sampler (e.g. a pretrained model) or a dataset (in which case you'll get some peaky behaviors). It would be good for the authors to expand on this empirically.

Relatedly, this method depends on a softmax of a potentially peaky value to learn with an off-policy objective. If the prior is flat enough and the energy peaky enough, this may very well mean that most of the data is effectively thrown away and most compute wasted because its weight is 0. There are other methods (although the ones I think of would count as concurrent work, so I'm not holding it against the authors) that learn similar amortized inference policies $q \propto p(x) \exp(-\beta E[x])$ that are able to sample on-$q$-policy (some are also able to do both off-policy and on-policy), therefore potentially wasting less data. Again not holding it against the authors to compare to those methods, but I'd love to see a deeper empirical analysis of when the proposed method works with respect to different energy landscapes; this could inform someone picking a method for their problem (and it is likely that the proposed method would be a top choice for certain problems, but not others).

Finally, I wonder what is the effect of estimating an expectation with a minibatch rather than a more precise method. For example, in the offline RL case, assuming all trajectories in the dataset come from the same policy $\mu$, then it would be much more precise to estimate $\mathbb{E}_\mu[Q^\psi]$ once over the dataset (especially if $\psi$ rarely changes), but maybe this doesn't change much.

---

> ### Author Response · Authors · 2024-11-20
> **Response to Reviewer 6Qqj (1/2)**
>
> Thank you for your strong support for our paper! We answered your questions as follows:
>
> **Q1. How is the variance introduced by the importance sampling?**
>
> **A1.** We acknowledge that using importance sampling to sample from $q(x) \propto p(x) \exp(E(x))$ [1, 2] might introduce additional variance. However, since we only use importance sampling to construct the loss function, we have found empirically that the negative impact of this variance is minimal. Moreover, we want to highlight that the batch size we use for training is relatively small and consistent with the baseline algorithms (64 for both image synthesis and molecular structure generation). Therefore, we do not need to increase the batch size specifically to control variance.
>
> **Q2. What are the trade offs that would inform choosing this method over others?**
>
> **A2.** We highlight the trade offs that would inform choosing our methods over the others as follows: our method is the first to directly learn exact energy-guided flow matching or diffusion models without relying on auxiliary models. In contrast, CEP [3] not only requires importance sampling to train its intermediate energy function $E_t(x)$  through contrastive learning (as shown in Eq. (13) of [3]), but it also necessitates additional back-propagation on its auxiliary model $E_t(x)$ . Moreover, methods like CG [4] and CFG [5] lack guarantees for generating the exact distribution, as discussed in Section 4.3.
>
> **Q3. Can this method be extended to more general tasks?**
>
> **A3.** We appreciate your insights on the potential extensions of our algorithm beyond RL. In response, we conducted experiments on image synthesis and molecular generation, following established benchmarks (DDPM [6], EDM [7]). Due to time constraints during the rebuttal period, we have included preliminary results in **Appendix F.3** and **F.4**. These results indicate that energy-weighted diffusion can be readily applied to guide complex generative tasks, where standard RL algorithms may not be effective.
>
> **Q4. What’s the performance of the algorithm for a different energy landscape?**
>
> **A4.** We acknowledge that when the energy function is highly peaky, it becomes challenging to learn and sample effectively. This challenge is well-known in fields like non-convex optimization and Langevin dynamics. In response to your question, we modified the energy function in the 8-Gaussian example and presented the results in **Appendix F.1**. The results indicate that even with a changed energy landscape, the energy-weighted method is still able to generate samples effectively. Additionally, we note that the energy function used in the existing "chessboard" examples (Figures 2 and 3) is also non-convex with four distinct peaks.
>
> In summary, while we acknowledge that learning an “extremely” peaky energy function is inherently difficult, our energy-weighted methods have demonstrated robust performance across different energy landscapes, including challenging ones.
>
> [1] Cremer, Julian, et al. "Multi-Objective Guidance via Importance Sampling for Target-Aware Diffusion-based De Novo Ligand Generation." ICML'24 Workshop ML for Life and Material Science: From Theory to Industry Applications.
>
> [2] Chen, Huayu, et al. "Offline Reinforcement Learning via High-Fidelity Generative Behavior Modeling." The Eleventh International Conference on Learning Representations.
>
> [3] Lu, Cheng, et al. "Contrastive energy prediction for exact energy-guided diffusion sampling in offline reinforcement learning." International Conference on Machine Learning. PMLR, 2023.
>
> [4] Dhariwal, Prafulla, and Alexander Nichol. "Diffusion models beat gans on image synthesis." Advances in neural information processing systems 34 (2021): 8780-8794.
>
> [5] Ho, Jonathan, and Tim Salimans. "Classifier-free diffusion guidance." arXiv preprint arXiv:2207.12598 (2022).
>
> [6] Ho, Jonathan, Ajay Jain, and Pieter Abbeel. "Denoising diffusion probabilistic models." Advances in neural information processing systems 33 (2020): 6840-6851.
>
> [7] Hoogeboom, Emiel, et al. "Equivariant diffusion for molecule generation in 3d." International conference on machine learning. PMLR, 2022.

---

> ### Author Response · Authors · 2024-11-20
> **Response to Reviewer 6Qqj (2/2)**
>
> **Q5. What if the data distribution is not the same as the prior distribution?**
>
> **A5**. Similar to diffusion models, our method assumes that the prior distribution matches the data distribution. We acknowledge that the algorithm’s performance may be impacted by irregular data present in the dataset or from a pre-trained model. There is a body of work [5, 6] aimed at filtering such irregular data to improve training outcomes. However, we emphasize that addressing irregular data is not the primary focus of this work. We are confident that these existing methods can be integrated into our framework to handle this challenge effectively.
>
>
> **Q6. What is the effect of estimating an expectation with a minibatch rather than a universal constant across the dataset?**
>
> **A6.** We acknowledge that using a global $\mathbb{E}_{p(x)}[\exp(\beta E(x))]$ as a constant could potentially reduce the variance compared to estimating the expectation over a minibatch. However, our choice to estimate the expectation using a minibatch was made to ensure that the proposed algorithm remains flexible and easily adaptable to different tasks, with minimal hyperparameter tuning or dataset preprocessing. Empirically, we have observed that the behavior of these two methods is quite similar, suggesting that the impact of estimating the expectation using a minibatch is not significant in practice.
>
> [8] Lin, Guang, et al. "Robust Diffusion Models for Adversarial Purification." arXiv preprint arXiv:2403.16067 (2024).
>
> [9] Xiao, Chaowei, et al. "Densepure: Understanding diffusion models for adversarial robustness." The Eleventh International Conference on Learning Representations. 2023.

---

> > ### Author Response · Authors · 2024-11-22
> > **Follow up with Reviewer 6Qqj**
> >
> > Thank you once again for your support of our paper! We’d like to follow up to ensure that our response has addressed your questions and concerns. Specifically, we mainly:
> > - Provided experiments on image synthesis tasks (Appendix F.3) and molecular generation tasks (Appendix F.4).
> > - Discussed the energy landscape and conducted an ablation study (Appendix F.2).
> > - Examined the details of importance sampling and its potential impact.
> >
> > In addition, we’ve uploaded a revised version with more experiments to enhance clarity. We hope this addresses your concerns and look forward to further discussion to answer any remaining questions you might have.

---

> > > ### Comment · Reviewer_6Qqj · 2024-11-26
> > > **Reply to Authors**
> > >
> > > Thank you for all the follow ups, the additional experiments are a neat addition.
> > >
> > > >  our method is the first to directly learn exact energy-guided flow matching or diffusion models without relying on auxiliary models
> > >
> > > There's a recent line of work generative flow networks that are also technically exact energy-guided diffusion models, may be worth a look at [1].
> > >
> > > [1] Improved off-policy training of diffusion samplers, Marcin Sendera, Minsu Kim, Sarthak Mittal, Pablo Lemos, Luca Scimeca, Jarrid Rector-Brooks, Alexandre Adam, Yoshua Bengio, Nikolay Malkin, NeurIPS 2024

---

> ### Author Response · Authors · 2024-11-26
> **Thank you for your feedback!**
>
> Thank you for your feedback and for acknowledging our response! We appreciate you pointing out a recent related work. We have included it as part of the discussion on related sampling methods in the revision. Additionally, we would like to provide a more detailed comparison here.
>
> Specifically, [1] addresses sampling from the Boltzmann distribution $p_{\text{target}}$ using a tractable initial distribution $\mu$ in Section 3, and it explored credit assignment via the energy function gradient  $\nabla_x \mathcal{E}(x)$ in Section 4.1. In comparison, [1] requires computing the gradient $\nabla_x \mathcal{E}(x)$ through back-propagation for the inductive bias. Therefore it will be slower when evaluating $u(x)$  as defined in Equation (14), similar with the CEP [2]. In our QIPO algorithm, the $u(x)$ is directly learnt from the Q-weighted loss function so the evaluation time should be significantly reduced (see Table 3). Therefore we claimed that our method is the first that directly (without any additional gradient requirements) learn the energy-guided flow.
>
> We appreciate your comment again and hope our response have addressed all your questions!
>
> [1] Improved off-policy training of diffusion samplers, Marcin Sendera, Minsu Kim, Sarthak Mittal, Pablo Lemos, Luca Scimeca, Jarrid Rector-Brooks, Alexandre Adam, Yoshua Bengio, Nikolay Malkin, NeurIPS 2024
>
> [2] Lu, Cheng, et al. "Contrastive energy prediction for exact energy-guided diffusion sampling in offline reinforcement learning." International Conference on Machine Learning. PMLR, 2023.

---

### Official Review · Reviewer_MVYW · 2024-11-04

**Soundness:** 3
**Presentation:** 3
**Contribution:** 1
**Rating:** 5
**Confidence:** 5

**Summary:**

This paper introduces the energy-based flow matching to offline reinforcement learning without the need for auxiliary models, and then propose a novel method called Q-weighted Iterative Policy Optimization based on this framework.

**Strengths:**

1.	The manuscript is generally well-written and well-organized.
2.	The paper presents comprehensive theoretical proof.

**Weaknesses:**

1.	In the reviewer’s opinion, the authors missed some existing works including EDP, and QVPO [1, 2], which also train the diffusion policy with the weighted loss. In that case, the paper does not propose anything particularly novel.
2.	Given weakness 1, the authors overclaim their contribution: “Our algorithm is the first energy-guided diffusion model that operates independently of auxiliary models and the first exact energy-guided flow matching model in the literature”.
3.	The reviewer thinks the proposed method should not be viewed as an energy-guided diffusion model but as an extension of RWR (reward-weighted regression) [3] via flow matching.
4.	Results in Table 2 does not show a distinct superiority of the proposed QIPO compared with previous methods.
5.	The motivation for applying flow matching to offline RL is not clear. It seems the multimodality of diffusion policy cannot be obviously improved with flow matching loss compared with normal diffusion loss.

[1] Kang B, Ma X, Du C, et al. Efficient diffusion policies for offline reinforcement learning[J]. Advances in Neural Information Processing Systems, 2024, 36.

[2] Ding S, Hu K, Zhang Z, et al. Diffusion-based Reinforcement Learning via Q-weighted Variational Policy Optimization[J]. arXiv preprint arXiv:2405.16173, 2024.

[3] Peters J, Schaal S. Reinforcement learning by reward-weighted regression for operational space control[C]//Proceedings of the 24th international conference on Machine learning. 2007: 745-750.

**Questions:**

1.	What is the motivation of applying flow matching to offline RL?
2.	Compared with the previous works for diffusion policy with weighted loss, what is the real contribution of this paper?

---

> ### Author Response · Authors · 2024-11-20
> **Response to Reviewer MVYW (1/2)**
>
> Thank you for your feedback. We address your questions below.
>
> **Q1. What is the contribution of this paper compared with reward-weighted regression EDP and QVPO?**
>
> **A1.** Thank you for pointing out these references. We have included a detailed discussion of these papers in **Appendix A.** of the revised manuscript. Although the *energy-weighted* flow matching (and diffusion) loss appears similar to the *reward-weighted regression* in EDP or QVPO, they are fundamentally different for several reasons:
> - **Motivation**: The energy-weighted diffusion model is derived from controlling the generation process of $q(x) \propto p(x) \exp(-\beta E(x))$, where both the original distribution $p(x)$ and the energy function $\beta E(x)$ are equally important. On the other hand, reward-weighted (policy) regression is formulated as minimizing the loss $\mathbb{E}[f(Q(s, a)) \log \pi_{\theta}(a | s)]$, as discussed in EDP [1]. The primary focus in RL is to maximize the $Q$ function, which differs significantly from generative modeling tasks where the goal is to generate high-quality samples while maintaining low energy. This distinction separates RL objectives from generative modeling objectives.
> - **Theoretical Insights**: Building on our motivation, we provide theoretical guarantees that energy-weighted flow matching can accurately learn the probability distribution $q(x) \propto p(x) \exp(-\beta E(x))$ (in RL is $\pi(a | s) \propto \mu(a | s) \exp(\beta Q(s, a))$. In contrast, neither EDP [1] nor QVPO [2] can precisely learn this energy-guided distribution or provide such theoretical guarantees, as detailed below.
> - **Methodology**: Although weighted regression is commonly used in literature, the Energy-Weighted diffusion differs from EDP and QVPO and provide unique perceptive:
> 	- Energy-Weighted Diffusion (ours) uses weights of the form $\exp(\beta Q(s, a)) / \mathbb{E}_{a \in \mu(\cdot | s)} \exp(\beta Q(s, a))$, which are guaranteed to generate $q(x) \propto p(x) \exp(-\beta E(x))$.
> 	- EDP uses a more artifically defined weight, $f(Q(s, a))$. Specifically,
> 	    - $f_{\text{CRR}}(Q(s, a)) = \exp\left(\beta Q(s, a) - \mathbb{E}_{a \in \hat{\pi}} Q(s, a)\right)$, where $\hat{\pi} = \mathcal{N}(\hat{a}_0, I)$ and $\hat{a}_0$ is evaluated using the score network. This formulation oversimplifies the behavioral policy as a Gaussian distribution, making it different from the exact energy-guided distribution. Our approach, in contrast, directly samples from the behavioral policy $\mu(a | s)$.
>         - $f_{\text{IQL}}(Q(s, a)) = \exp(Q(s, a) - V(s))$ suffers from a similar issue, as it replaces $\mathbb{E}_{a \in \hat{\pi}} Q(s, a)$ with $V(s)$. This difference prevents it from achieving the exact energy-guided distribution, and IQL also requires a pre-trained value function.
>     - QVPO employs a linear weight $Q(s, a)$. Its goal, as discussed by the authors, is to encourage exploration in online RL, where exponential weighting is more suitable for offline RL.
>     - As the theoretical analysis presented in Theorem 3.3 and Theorem 4.3, neither EDP nor QVPO can strictly recover the distribution $q(x) \propto p(x) \exp(-\beta E(x))$. While these algorithms improve $Q$ values, this is not the sole objective in generative modeling.
> - **Objective**: Our methods balance alignment with the original policy $p(x)$ and energy guidance $E(x)$. This balance makes our approach applicable to diverse tasks, such as image synthesis and molecular structure generation, as demonstrated in **Appendix F.3** and **F.4** of the revision. In these tasks, simply maximizing the reward (e.g., generating a pure blue image) is inadequate; the generative model must align with the underlying dataset distribution.
> - **Empirical Performance**: In comparison with EDP in the offline RL setting, our algorithm achieves superior performance, as presented in the table below. This highlights the importance of the theoretical guarantee on the distribution and the contribution of our paper.
>
> |Task | QIPO (ours) | EDP + TD3+BC | EDP + CRR | EDP + IQL|
> | ---- | ---- | ---- | ---- | ---- |
> | Locomotion (Average) | **86.2**| 85.5 | 78.3 | 79.9|
> | Antmaze (Average) | **77.3** | 29.8 | 43.8 | 73.4|
>
> [1] Kang B, Ma X, Du C, et al. Efficient diffusion policies for offline reinforcement learning[J]. Advances in Neural Information Processing Systems, 2024, 36.
>
> [2] Ding S, Hu K, Zhang Z, et al. Diffusion-based Reinforcement Learning via Q-weighted Variational Policy Optimization[J]. arXiv preprint arXiv:2405.16173, 2024.
>
> [3] Peters J, Schaal S. Reinforcement learning by reward-weighted regression for operational space control[C]//Proceedings of the 24th international conference on Machine learning. 2007: 745-750.

---

> ### Author Response · Authors · 2024-11-20
> **Response to Reviewer MVYW (2/2)**
>
> **Q2. What’s the motivation for introducing flow matching to RL?**
>
> **A2.** Flow matching serves as a generalized framework for diffusion models, offering several advantages, including simulation-free implementation and faster convergence compared to diffusion models. These benefits have been demonstrated in our experiments (see Table 3 in our paper). We believe that flow matching provides **a more general framework** for studying the generative models in RL and also provides **a faster convergence rate** when generating new actions.
>
> Additionally, we want to emphasize that flow matching has been successfully developed in various fields, such as image synthesis [4] and molecular structure generation [5]. Notably, the recent work on $\mathbf{\pi_0}$ general robotics control [6] utilizes flow matching to manage real-world robotic tasks. We believe this example illustrates the practical significance of flow matching in RL, as it demonstrates the potential to improve control and performance in complex environments.
>
> [4] Lipman, Yaron, et al. "Flow Matching for Generative Modeling." The Eleventh International Conference on Learning Representations.
> [5] Song, Yuxuan, et al. "Equivariant flow matching with hybrid probability transport for 3d molecule generation." Advances in Neural Information Processing Systems 36 (2024).
> [6] Black, Kevin, et al. "$\pi_0 $: A Vision-Language-Action Flow Model for General Robot Control." arXiv preprint arXiv:2410.24164 (2024).

---

> > ### Author Response · Authors · 2024-11-22
> > **Follow up with Reviewer MVYW**
> >
> > Thank you again for your questions and feedback. We’d like to follow up to see if you have any additional questions after reviewing our response. Specifically, in our previous communication, we:
> >
> > - Highlighted the novelty of our contribution in understanding the energy-guided distribution model, which involves not only maximizing the reward but also aligning with the original distribution.
> > - Emphasized that aligning with the original distribution is a crucial aspect of generative modeling. We strongly believe that our method focuses on energy-guided flow matching rather than being an extension of RWR. While these methods may appear similar at first glance, they target fundamentally different objectives.
> > - Demonstrated significant performance improvements using the approaches you mentioned. We believe our algorithm achieves state-of-the-art (SOTA) performance with a deeper conceptual understanding and simpler implementation.
> > - Provided further explanations on flow matching and its applications in RL and robotic tasks.
> >
> > Additionally, we’ve uploaded a revision with more experiments, extending the proposed algorithm to image synthesis (Appendix F.3) and molecular generation (Appendix F.4). We hope this provides additional clarity and invite further discussion to address any remaining questions you might have.

---

> > > ### Comment · Reviewer_MVYW · 2024-11-24
> > > **Reply to authors**
> > >
> > > Thanks for your feedback. However, the reviewer still considers that this paper's novelty is limited. The proposed weight function just adds a normalization term in the denominator compared with EDP. Actually, this normalization is inappropriate in some sense. The Q or advantage value in a state can reflect the importance of "making right decision" in this state. Thus, this normalization term in the denominator will destroy this preference of policy model for focusing on "important states".
> > >
> > > Moreover, the reviewer truly suggests the authors introduce previous works based on weight-based optimization and diffusion models in related works rather than in appendix since the idea of this paper is quite similar to theirs.
> > >
> > > Finally, the authors claim QIPO has a faster convergence rate, but do not provide any theoretical analysis on the convergence rate or experimental results (e.g., learning curves) to confirm it. However, given the completeness of this paper, the reviewer is willing to raise the score if the authors can provide more analysis or experiments to clarify the advantages of applying flow matching to offline RL.

---

> > > > ### Author Response · Authors · 2024-11-24
> > > > **Follow up with Reviewer MVYW (1/2)**
> > > >
> > > > Thank you for your feedback. We appreciate your willingness to consider raising the score. We have also updated a revison providng the 1) training curves for OT and diffusion (**Appendix F.5**) and 2) some highlights on other weighted regression works (**Section 5**). Below, we address your questions in detail:
> > > >
> > > > **Q1. What is the novelty of this paper compared to EDP?**
> > > >
> > > > **A1.** We would like to reiterate that the primary contribution of this paper lies in proposing an energy-weighted loss to correctly learn the target distribution $q(x) \propto p(x) \exp(E(x))$ . While we acknowledge that weighted regression techniques have been widely used in reinforcement learning (RL), our work is the first to explicitly establish the connection between energy-guided target distributions ($q$) and energy-weighted training losses. Moreover, the formulation and theoretical understanding of this energy-weighted process enable iterative policy improvement, as demonstrated in Equation (5.4). We believe that both the theoretical analysis and the iterative approach introduced here represent a novel contribution to the literature.
> > > >
> > > >
> > > > **Q2. Can the authors add more related works on weight-based optimization?**
> > > >
> > > > **A2.** Thank you for the suggestion. In the revision, we will incorporate additional related works by moving **Appendix A** to the main text and expanding the discussion to address the current page limit. Furthermore, though the overall scope centers on the energy-guided generative model, we will add more remarks and discussions in **Section 5** when introducing the methodology in the context of offline RL. In our newest revision we have added one sentense reflecting this update.

---

> > > > > ### Author Response · Authors · 2024-11-24
> > > > > **Follow up with Reviewer MVYW (2/2)**
> > > > >
> > > > > **Q3. Can the authors provide more analysis or experiments to clarify the advantages of flow matching**
> > > > >
> > > > > Thank you for the question. We aim to demonstrate the advantages of flow matching (specifically Optimal Transport, OT) from both theoretical and empirical perspectives:
> > > > >
> > > > >
> > > > > **Theoretical Analysis.**
> > > > >
> > > > > - **Constant Direction of the Vector Fields.** Prior works [1, 2] have shown that OT provides a vector field with constant direction over time [1]. Specifically, the OT vector field is given by  $u(x | x_0) = \frac{x_0 - (1 - \sigma_{\min})\epsilon}{1 - (1 - \sigma_{\min})t}$ , thus the direction of $u(x | x_0)$ remains unchanged with respect to $t$ . This leads to a simpler regression task, as discussed in [1] (Sec. 4, Example II).
> > > > > - **Constant Speed of Particle Movement.** OT and diffusion processes utilize different schedules $(\mu_t, \sigma_t)$ to add noise. For instance, in DDPM [3] or score matching [4], $\mu_t$ is defined as $\mu_t = \exp(-0.5\beta_0 t - 0.25 \beta_1 t^2)$, and $\sigma_t = \sqrt{1 - \mu_t^2}$. This exponential decay causes the particle movement to slow as $t \to 1$ but to accelerate too rapidly (potentially “overshooting”) as  $t \to 0$, as noted in [1] (Sec. 4). In contrast, OT employs $\mu_t = 1 - t$ and $\sigma_t = 1 - (1 - \sigma_{\min})t$, ensuring particles move at a constant speed from the initial Gaussian distribution to the target distribution. This implies that the reverse process of OT can be more easily accelerated compared to diffusion.
> > > > >
> > > > > **Empirical Evaluation.**
> > > > >
> > > > > - **Training Loss Curve.** We present the training loss in Appendix F.5. Both diffusion and OT are trained for 600 epochs to ensure a fair comparison. As shown in Figure F.3, the OT loss converges to its final value slightly faster than diffusion. This demonstrates that OT achieves (slightly) faster convergence during training.
> > > > > - **Evaluation Time.** Most importantly, one of the key priorities in RL is efficient sampling from the policy $\pi$. We report the time consumption for this in Table 3 and Appendix E.2. Both diffusion and OT utilize DPM-solver [5] with identical parameters to accelerate the reverse process. However, OT achieves an average speed-up of approximately 50% over diffusion (**27.26ms** (OT) vs. 55.86ms (diffusion)). This observation is consistent with our Theoretical Analysis 2.
> > > > >
> > > > > **Conclusion**
> > > > > While we have demonstrated the advantages of flow matching over diffusion—particularly in time efficiency for sampling from  \pi —we wish to emphasize that the **main contribution** of this paper is still on the **energy-guidance** in general flow matching framework. The proposed algorithm is practical and novel in both flow matching and regular diffusion with demonstrated theoretical insights.
> > > > >
> > > > > [1] Lipman, Yaron, et al. "Flow Matching for Generative Modeling." The Eleventh International Conference on Learning Representations.
> > > > >
> > > > > [2] Zheng, Qinqing, et al. "Guided flows for generative modeling and decision making." arXiv preprint arXiv:2311.13443 (2023).
> > > > >
> > > > > [3] Ho, Jonathan, Ajay Jain, and Pieter Abbeel. "Denoising diffusion probabilistic models." Advances in neural information processing systems 33 (2020): 6840-6851.
> > > > >
> > > > > [4] Song, Yang, et al. "Score-based generative modeling through stochastic differential equations." arXiv preprint arXiv:2011.13456 (2020).
> > > > >
> > > > > [5] Lu, Cheng, et al. "Dpm-solver: A fast ode solver for diffusion probabilistic model sampling in around 10 steps." Advances in Neural Information Processing Systems 35 (2022): 5775-5787.

---

### Author Response · Authors · 2024-11-20
**Response to all reviewers**

We appreciate all reviewers for their valuable feedback and for recognizing the significance of our work in guided generation, as well as the contributions of our algorithm.

We have uploaded a revised version that includes additional discussions and experiments. The revisions are highlighted in red. The key changes are summarized below:

- **In response to Reviewer MVYW**: We added **Appendix A** to elaborate on the differences between the energy-guided generative model and reward-weighted regression. In summary, energy guidance serves for generative models not only for maximizing rewards but also for aligning with the data distribution.
- **In response to Reviewer 6Qqj**: We added **Appendix F.1** to present experiments with a modified energy landscape for the 8-Gaussian distribution, making it more “peaky.” We also included discussions to show that the energy-weighted diffusion model can still accurately generate the target distribution.
- **In response to Reviewer JYA4**: We included an algorithm in **Appendix F.2** to demonstrate how the score function can take the guidance scale $\beta$ as an input. We explain that $\beta$ can be treated similarly to time $t$, and we conducted experiments on bandit tasks to verify the algorithm’s performance.
- **In response to Reviewers mBzj and MVYW**: We added new tasks on image synthesis and molecular structure generation, described in **Appendix F.3** and **Appendix F.4**. Due to time constraints during the author response period, we had to limit the image resolution to 128 x 128 and reduce the training iterations for the molecular structure generation. Nevertheless, our results demonstrate that the proposed methods effectively guide the diffusion model toward the desired distribution.
- In response to Reviewer mBzj: We corrected a typo in the proof of Theorem 4.1 in **Appendix B.1**.

We have also addressed each comment in the individual rebuttals. We hope the revisions address all concerns and questions, and we are happy to discuss any further feedback from the reviewers!

---

### Meta-Review · Area_Chair_G5G4 · 2024-12-20

**Metareview:**

Thank you for your submission to ICLR. This paper studies energy guidance in flow matching, and introduces the procedure energy-weighted flow matching (EFM), which, in contrast with prior work, does not require auxiliary models. The authors also present energy-weighted diffusion models. They apply these methods to tasks in offline reinforcement learning, where they shows strong performance.

The reviewers are generally positive about the merits of this paper, the benefits of the proposed method, the soundness of the theory, and downstream use for offline reinforcement learning. They also appreciate that this paper is written clearly, and the algorithm is intuitive. The authors have followed up with a number of additional results, which improve the quality of the paper. I thus recommend this paper for acceptance.

**Additional Comments On Reviewer Discussion:**

After rebuttal and discussion, a couple of reviewers felt that their questions were sufficiently answered and raised their scores. All of the reviewers remained mostly positive about this paper. We encourage the authors to follow through on their agreement to add requested prior work (e.g. on weight-based optimization and diffusion models) to the Related Work section of their paper.

---

### Decision · Program_Chairs · 2025-01-22

Accept (Poster)